# Efficient Difference-in-Differences Estimation when Outcomes are Missing at Random

## Abstract

The Difference-in-Differences (DiD) method is a fundamental tool for causal inference, yet its application is often complicated by missing data. Although recent work has developed robust DiD estimators for complex settings like staggered treatment adoption, these methods typically assume complete data and fail to address the critical challenge of outcomes that are missing at random (MAR) – a common problem that invalidates standard estimators. We develop a rigorous framework, rooted in semiparametric theory, for identifying and efficiently estimating the Average Treatment Effect on the Treated (ATT) when either pre- or post-treatment (or both) outcomes are missing at random. We first establish nonparametric identification of the ATT under two minimal sets of sufficient conditions. For each, we derive the semiparametric efficiency bound, which provides a formal benchmark for asymptotic optimality. We then propose novel estimators that are asymptotically efficient, achieving this theoretical bound. A key feature of our estimators is their multiple robustness, which ensures consistency even if some nuisance function models are misspecified. We validate the properties of our estimators and showcase their broad applicability through an extensive simulation study.

## 1 Introduction

The Difference-in-Differences (DiD) method stands as a cornerstone of modern applied research in economics (Lechner et al., 2011), social sciences (Greene & Liu, 2021), and public health (Wing et al., 2018), prized for its intuitive and powerful approach to estimating the causal effects of policies and interventions from *observational panel*, or *longitudinal*, data. In its canonical form, the method is used to estimate the *average treatment effect on the treated* (ATT), by leveraging a core identifying condition, the *parallel trends assumption*. This assumption posits that, had the treatment not occurred, the outcomes for the treated and control groups would have followed similar (parallel) trajectories. This core assumption allows researchers to use the observed change in the control group's outcome to construct the counterfactual path of the treated group, thereby isolating the treatment's causal impact by differencing out confounding factors that are constant over time.

While the classic two-group, two-period setup provides a clear theoretical foundation, empirical applications are rarely so simple. The last decade has seen a critical reexamination of how DiD methods are applied in more complex settings – see Callaway (2023); de Chaisemartin & D'Haultfœuille (2023); Roth et al. (2023) for a review. A major focus has been on scenarios with multiple time periods and variation in when different units receive treatment, known as "staggered adoption" (Callaway & Sant'Anna, 2021; De Chaisemartin & d'Haultfoeuille, 2020; Goodman-Bacon, 2021; Sun & Abraham, 2021). A second parallel line of inquiry has developed methods to handle missingness in post-treatment outcomes, often under stringent assumptions. For example, Rathnayake et al. (2024); Shin (2024); Viviens (2025) provide estimation strategies for the ATT in an unconditional framework where available covariates are disregarded; while Bellégo et al. (2025), working in a staggered adoption setting, assumes that outcomes are observed at least twice for each unit. However, despite the progress made, none of these papers discusses efficiency in either nonparametric or semiparametric models in two-group, two-period DiD setups with available covariates. A further persistent challenge, commonly encountered in practice, is how to conduct DiD analysis when *pre-treatment* outcome data are missing for a subset of the sample. This situation is common in many real scenarios: in labor economics, individuals entering a job training program may lack pre-

treatment earnings records; in education policy, students may transfer into a school district without their prior test scores; and in health research, electronic health records may not contain a baseline measurement for patients who did not have a clinical visit during the pre-period. When the pre-treatment outcome is unavailable for some units, the standard DiD estimator cannot be computed. A simple "complete-case" analysis that discards observations with missing data can be deeply flawed, as it may introduce significant selection bias. In fact, if the mechanism that causes the data to be missing is related to treatment or other characteristics that influence the outcome, the remaining sample is no longer representative of the population of interest and the resulting estimates will be biased.

This paper addresses this critical gap by bridging modern DiD estimation with semiparametric statistical theory (Bickel et al., 1993; Kennedy, 2024; Tsiatis, 2006). We provide a formal framework for identifying and efficiently estimating the ATT when pre-treatment outcomes are *missing at random* (MAR). In the Appendix, we also extend our results to the symmetric scenario where all pre-treatment outcomes are observed, but post-treatment outcomes can be missing at random. Finally, again in the Appendix, we also further extend our framework to accommodate the scenario in which outcomes can be missing both before and after treatment.

## 1.1 PROBLEM SETUP

We assume a two-group, two-period DiD setup[1]. In particular, we assume to have access to a collection of $n$ i.i.d. observed-data samples $\mathcal{D}_i = (X_i, R_{i0}, R_{i0}Y_{i0}, A_i, Y_{i1}) \sim \mathbb{P}^\star$, $i = 1, \ldots, n$, where $X_i \in \mathbb{R}^p$ is a $p$-dimensional vector of pre-treatment covariates; $R_{i0} \in \{0, 1\}$ is a binary variable indicating whether the baseline, pre-treatment outcome $Y_{i0}$ is observed ($R_{i0} = 1$) or not ($R_{i0} = 0$); $A_i \in \{0, 1\}$ is a binary variable indicating whether observation $i$ has received a treatment ($A_i = 1$) or was in the control group ($A_i = 0$); finally $Y_{i1} = A_i Y_{i1}^{(1)} + (1 - A_i)Y_{i1}^{(0)}$ is the observed outcome after treatment, where $Y_{i1}^{(1)}$ and $Y_{i1}^{(0)}$ denote the potential outcomes for observation $i$ under treatment and control, respectively. We let $\mathcal{D} = (X, R_0, R_0Y_0, A, Y_1)$, with $Y_1 = AY_1^{(1)} + (1 - A)Y_1^{(0)}$, denote an independent copy of $\mathcal{D}_i$. From a theoretical viewpoint, it is also useful to construct a prototypical full-data sample as $\mathcal{D}^F = (X, Y_0, Y_1^{(0)}, Y_1^{(1)})$ – see Tsiatis (2006).

**Remark 1.1.** We want to emphasize that our framework differs from both the balanced panel data and the repeated cross-section data analyzed by Sant'Anna & Zhao (2020). The former postulates that each sample is observed both before and after treatment: the latter assumes that each sample is observed either before or after treatment. Instead, our setup mirrors the so-called *unbalanced*, or partially missing, panel data framework, where the outcomes of some samples are observed both before and after the treatment, while some other samples miss pre-treatment outcomes.

Our goal is the estimation of the *average treatment effect on the treated* (ATT), defined as:

$$\theta^\star = \text{ATT} = \mathbb{E}\left[Y_1^{(1)} - Y_1^{(0)} \mid A = 1\right]$$
$$= \mathbb{E}\left[Y_1^{(1)} - Y_0 \mid A = 1\right] - \mathbb{E}\left[Y_1^{(0)} - Y_0 \mid A = 1\right]. \tag{1}$$

The ATT is a function of the full data $\mathcal{D}^F$, and as such is not directly identifiable using only observed data $\mathcal{D}$. In the next Section, we provide a minimal set of assumptions that make the target $\theta^\star$ a function of the observed data $\mathcal{D}$.

## 2 IDENTIFICATION AND EFFICIENCY BOUNDS

We provide a first set of assumptions for identifiability.

**Assumption 2.1** (Identifiability). Let the following identifiability assumptions hold:

    **a. Conditional parallel trends.** $Y_1^{(0)} - Y_0 \perp\!\!\!\perp A \mid X$.

    **b. Consistency.** $Y_1 = AY_1^{(1)} + (1 - A)Y_1^{(0)}$.

---

[1]The framework can be extended to staggered adoption settings as in Callaway & Sant'Anna (2021), but we work in a two-group, two-period setting for ease of exposition.

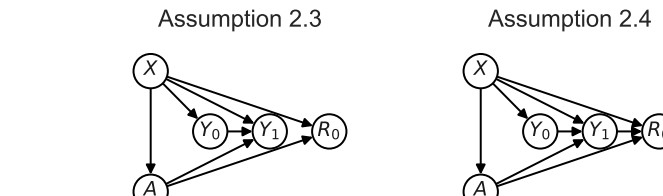

Figure 1: Example DAGs showing some of the causal dependencies allowed under Assumptions 2.3 and 2.4.

   **c. Positivity.** $\pi^\star(x) = \mathbb{P}\left[A = 1 \mid X = x\right] \in (0, 1)$ almost surely for every $x \in \mathbb{R}^p$.

**Remark 2.2.** Assumption **a** is stronger than the conditional mean assumption standardly employed in the econometrics literature on DiD usually expressed as:

$$\mathbb{E}\left[Y_1^{(0)} - Y_0 \mid X, A = 1\right] = \mathbb{E}\left[Y_1^{(0)} - Y_0 \mid X, A = 0\right] . \tag{2}$$

We use the stronger conditional independence notation in Assumption **a** for "notational consistency" because it aligns with the standard conditional independence language used in semiparametric theory and in our subsequent MAR assumptions. Importantly, Assumption **a** does not imply conditional independence of either $Y_1^{(0)}$ or $Y_0$, but only of their incremental difference. Assumptions **b** and **c** are standard in causal inference. In particular, Assumption **b** guarantees the observed outcome at time $t = 1$ corresponds to the potential outcome associated with the treatment status. Assumption **c** forces the probability of being treated, for each observation, to be strictly positive.

Then, we also provide two different sets of assumptions for the missingness mechanism in the pre-treatment outcome.

**Assumption 2.3** (Outcome independent missing at random). Let the following MAR assumptions hold:

   **a. No unmeasured confounding.** $Y_0 \per\!\!\!\perp R_0 \mid X, A$.

   **b. Weak overlap.** $\gamma^\star(x, a) = \mathbb{P}\left[R_0 = 1 \mid X = x, A = a\right] \in (0, 1)$ almost surely for every $x \in \mathbb{R}^p$ and $a \in \{0, 1\}$.

**Assumption 2.4** (Outcome dependent missing at random). Let the following MAR assumptions hold:

   **a. No unmeasured confounding.** $Y_0 \per\!\!\!\perp R_0 \mid X, Y_1, A$.

   **b. Weak overlap.** $\gamma^\star(x, y_1, a) = \mathbb{P}\left[R_0 = 1 \mid X = x, Y_1 = y_1, A = a\right] \in (0, 1)$ almost surely for every $x \in \mathbb{R}^p$, $y_1 \in \mathbb{R}$, and $a \in \{0, 1\}$.

**Example 2.5** (Medical records motivation). To illustrate the practical distinction between these assumptions, consider a longitudinal study on the progression of a chronic disease, where $Y_0$ is the baseline health status and $Y_1$ is the post-treatment status. Under Assumption 2.3, missingness in baseline records ($R_0$) is driven solely by covariates ($X$) or treatment assignment ($A$). For instance, missingness might arise because a specific hospital system ($X$) used paper records that were not digitized, regardless of how sick the patients currently are. Under Assumption 2.4, missingness may be driven by the future outcome. For example, in retrospective chart reviews, the availability of baseline data ($R_0$) often depends on the patient's current condition ($Y_1$). Clinicians may be required to diligently hunt down and record historical data ($Y_0$) only for patients who currently exhibit severe complications ($Y_1$). Conversely, for patients who have recovered (low $Y_1$), the baseline charts may never be retrieved or digitized ($R_0 = 0$). In this setting, conditioning on $Y_1$ is necessary to block the dependence between missingness and the unobserved baseline outcome. See Figure 1 for a graphical representation using DAGs of the causal relationships under the two different assumptions.

**Remark 2.6.** These assumptions are novel to our setting and are equivalent to a *missing at random* (MAR) missingness design. Notice that we let the missingness pattern depend on both covariates $X$

*and* the treatment $A$, and eventually on the post-treatment outcome. In other words, we admit the possibility that pre-treatment outcomes can be missing due to covariates, the treatment that will be administered, and the post-treatment outcome values.

Equipped with the previous sets of assumptions, we can now identify the ATT as a function of the observed data $\mathcal{D}$ as shown in the following Lemma.

**Lemma 2.7** (Identification of ATT). *Under Assumption 2.1 and Assumption 2.3, the ATT can be identified as a function of the observed data:*

$$
\theta^\star = \mathbb{E}\left[\frac{A}{\mathbb{E}[A]}\left(Y_1 - \mathbb{E}[Y_0 \mid X, A = 1, R_0 = 1]\right)\right]
$$
$$
- \mathbb{E}\left[\frac{A}{\mathbb{E}[A]}\left(\mathbb{E}[Y_1 \mid X, A = 0] - \mathbb{E}[Y_0 \mid X, A = 0, R_0 = 1]\right)\right].
$$
(3)

*Under Assumption 2.1 and Assumption 2.4, the ATT can be identified as a function of the observed data:*

$$
\theta^\star = \mathbb{E}\left[\frac{A}{\mathbb{E}[A]}\left(Y_1 - \mathbb{E}[Y_0 \mid X, Y_1, A = 1, R_0 = 1]\right)\right]
$$
$$
- \mathbb{E}\left[\frac{A}{\mathbb{E}[A]}\left(\mathbb{E}[Y_1 \mid X, A = 0] - \mathbb{E}[\mathbb{E}[Y_0 \mid X, Y_1, A = 0, R_0 = 1] \mid X, A = 0]\right)\right].
$$
(4)

Once identified, the next natural question is how well the ATT can be estimated under each set of assumptions. Before proceeding, we denote the *regression functions* as $\mu_t^\star(x, a) = \mathbb{E}[Y_t \mid X = x, A = a]$ for $t \in \{0, 1\}$, and $\mu_0^\star(x, y_1, a) = \mathbb{E}[Y_0 \mid X = x, Y_1 = y_1, A = a]$. Furthermore, denote the *nested regression function* as $\eta_t^\star(x, a) = \mathbb{E}[\mu_t^\star(x, Y_1, a) \mid X = x, A = a]$. Notice that we cannot use the law of iterated expectations to simplify $\eta_t^\star(x, a)$ into $\mathbb{E}[Y_0 \mid X = x, A = a]$. The inner conditional expectation $\mu_t^\star(x, Y_1, a)$ is in fact implicitly conditioning on $R_0 = 1$, while the external conditional expectation is not. With this in mind, we suppress dependence of $\mu_t^\star(x, Y_1, a)$ on $R_0 = 1$ for notational simplicity. The following Proposition provides the semiparametric efficiency bounds.

**Proposition 2.8** (Semiparametric efficiency bounds). *Under Assumption 2.1 and Assumption 2.3, the efficient observed-data influence function is given by:*

$$
\varphi(\mathcal{D}) = \frac{A}{\mathbb{E}[A]}\left(Y_1 - \left(\mu_0^\star(X, 1) + \frac{R_0}{\gamma^\star(X, 1)}(Y_0 - \mu_0^\star(X, 1))\right) - \mu_1^\star(X, 0) + \mu_0^\star(X, 0)\right)
$$
$$
- \frac{(1-A)\pi^\star(X)}{(1 - \pi^\star(X))\mathbb{E}[A]}\left(Y_1 - \frac{R_0}{\gamma^\star(X, 0)}(Y_0 - \mu_0^\star(X, 0)) - \mu_1^\star(X, 0)\right) - \frac{A}{\mathbb{E}[A]}\theta^\star.
$$
(5)

*Under Assumption 2.1 and Assumption 2.4, the efficient observed-data influence function is given by:*

$$
\varphi(\mathcal{D}) = \frac{A}{\mathbb{E}[A]}\left(Y_1 - \left(\mu_0^\star(X, Y_1, 1) + \frac{R_0}{\gamma^\star(X, Y_1, 1)}(Y_0 - \mu_0^\star(X, Y_1, 1))\right) - \mu_1^\star(X, 0) + \eta_0^\star(X, 0)\right)
$$
$$
- \frac{(1-A)\pi^\star(X)}{(1 - \pi^\star(X))\mathbb{E}[A]}\left(Y_1 - \left(\mu_0^\star(X, Y_1, 0) + \frac{R_0}{\gamma^\star(X, Y_1, 0)}(Y_0 - \mu_0^\star(X, Y_1, 0))\right) - \mu_1^\star(X, 0) + \eta_0^\star(X, 0)\right)
$$
$$
- \frac{A}{\mathbb{E}[A]}\theta^\star.
$$
(6)

*The semiparametric efficiency bound under each assumption is given by* $\mathbb{V}[\varphi(\mathcal{D})] = \mathbb{E}[\varphi^2(\mathcal{D})]$.

**Remark 2.9.** When the pre-treatment outcome is always observed, we recover the results by Sant'Anna & Zhao (2020) (Proposition 1) and Hahn (1998) (Theorem 1). In fact, the efficient influence function in this case simplifies to

$$
\varphi^F(\mathcal{D}) = \frac{A}{\mathbb{E}[A]}\left(Y_1 - Y_0 - \mu_1^\star(X, 0) + \mu_0^\star(X, 0)\right)
$$
$$
- \frac{(1-A)\pi^\star(X)}{(1 - \pi^\star(X))\mathbb{E}[A]}\left(Y_1 - Y_0 - \mu_1^\star(X, 0) + \mu_0^\star(X, 0)\right) - \frac{A}{\mathbb{E}[A]}\theta^\star.
$$
(7)

Notice, however, that by allowing for missingness, our approach provides a better description of the loss in efficiency incurred when pre-treatment outcomes are missing at random. In particular, the efficiency loss incurred under Assumption 2.3 is:

$$
\begin{aligned}
\mathbb{V}\left[\varphi\left(\mathcal{D}\right)\right] - \mathbb{V}\left[\varphi^{F}\left(\mathcal{D}\right)\right] &= \mathbb{V}\left[\varphi\left(\mathcal{D}\right) - \varphi^{F}\left(\mathcal{D}\right)\right] \\
&= \mathbb{E}\left[\frac{\pi^{\star}(X)\mathbb{V}\left[Y_0 \mid X, A = 1\right]}{\mathbb{E}\left[A\right]^2}\frac{1 - \gamma^{\star}(X,1)}{\gamma^{\star}(X,1)}\right] \\
&\quad + \mathbb{E}\left[\frac{\pi^{\star}(X)^2\mathbb{V}\left[Y_0 \mid X, A = 0\right]}{(1 - \pi^{\star}(X))\mathbb{E}\left[A\right]^2}\frac{1 - \gamma^{\star}(X,0)}{\gamma^{\star}(X,0)}\right],
\end{aligned}
\tag{8}
$$

where we are exploiting the fact that the influence function without missingness is a projection of the observed-data influence function, and as such the variance of the observed-data influence functions decomposes nicely due to the Pythagorean theorem (Tsiatis, 2006). Similarly, the efficiency loss incurred under Assumption 2.4 is:

$$
\begin{aligned}
\mathbb{V}\left[\varphi\left(\mathcal{D}\right)\right] - \mathbb{V}\left[\varphi^{F}\left(\mathcal{D}\right)\right] &= \mathbb{E}\left[\frac{\pi^{\star}(X)\mathbb{V}\left[Y_0 \mid X, Y_1, A = 1\right]}{\mathbb{E}\left[A\right]^2}\frac{1 - \gamma^{\star}(X,Y_1,1)}{\gamma^{\star}(X,Y_1,1)}\right] \\
&\quad + \mathbb{E}\left[\frac{\pi^{\star}(X)^2\mathbb{V}\left[Y_0 \mid X, Y_1, A = 0\right]}{(1 - \pi^{\star}(X))\mathbb{E}\left[A\right]^2}\frac{1 - \gamma^{\star}(X,Y_1,0)}{\gamma^{\star}(X,Y_1,0)}\right] \\
&\quad + \mathbb{V}\left[\left(\frac{A}{\mathbb{E}\left[A\right]} - \frac{(1-A)\pi^{\star}(X)}{(1 - \pi^{\star}(X))\mathbb{E}\left[A\right]}\right)(\eta_0^{\star}(X,0) - \mu_0^{\star}(X,0))\right].
\end{aligned}
\tag{9}
$$

We now turn to the construction of estimators that can asymptotically match the semiparametric efficiency bound derived above.

## 3 ESTIMATION AND INFERENCE

The nuisance functions $\mu^{\star}$, $\pi^{\star}$, $\gamma^{\star}$, and $\eta^{\star}$ are unknown, and must be estimated from the data at hand. We employ *cross-fitting* to avoid restrictive Donsker conditions and to retain full-sample efficiency (Bickel & Ritov, 1988; Chernozhukov et al., 2018; Robins et al., 2008; Schick, 1986; Zheng & Van Der Laan, 2010). Cross-fitting works as follows. We first randomly split the observations $\{\mathcal{D}_1, \ldots, \mathcal{D}_n\}$ into $J$ disjoint folds (without loss of generality, we assume that the number of observations $n$ is divisible by $J$). For each $j = 1, \ldots, J$ we form $\hat{\mathbb{P}}^{[-j]}$ with all but the $j$-th fold, and $\mathbb{P}_n^{[j]}$ with the $j$-th fold. Then, we learn $\hat{\mu}^{[-j]}$, $\hat{\pi}^{[-j]}$, $\hat{\gamma}^{[-j]}$, and $\hat{\eta}^{[-j]}$ on $\hat{\mathbb{P}}^{[-j]}$, and compute the final estimator $\hat{\theta}$ by solving the estimating equation

$$
\sum_{j=1}^{J}\sum_{i \in \mathbb{P}_n^{[j]}}\varphi\left(\mathcal{D}_i; \theta^{\star}; \hat{\mu}^{[-j]}; \hat{\pi}^{[-j]}; \hat{\gamma}^{[-j]}; \hat{\eta}^{[-j]}\right) = 0.
\tag{10}
$$

For simplicity, we assume that $\hat{\eta}^{[-j]}$ is estimated on an independent subsample of $\hat{\mathbb{P}}^{[-j]}$, that is, it is independent from $\hat{\mu}^{[-j]}$, $\hat{\pi}^{[-j]}$, $\hat{\gamma}^{[-j]}$. The solution to the previous estimating equation can be more conveniently expressed as:

$$
\hat{\theta} = \frac{1}{J}\sum_{j=1}^{J}\hat{\theta}^{[j]},
\tag{11}
$$

where the form of $\hat{\theta}^{[j]}$ depends on the missing at random assumption. Under Assumption 2.1 and Assumption 2.3, it is equal to:

$$
\begin{aligned}
\hat{\theta}^{[j]} &= \sum_{i \in \mathbb{P}_n^{[j]}}\frac{A_i}{\sum_{i \in \mathbb{P}_n^{[j]}}A_i}\left(Y_{i1} - \left(\hat{\mu}_0^{[-j]}(X_i,1) + \frac{R_{i0}}{\hat{\gamma}^{[-j]}(X_i,1)}\left(Y_{i0} - \hat{\mu}_0^{[-j]}(X_i,1)\right)\right) - \hat{\mu}_1^{[-j]}(X_i,0) + \hat{\mu}_0^{[-j]}(X_i,0)\right) \\
&\quad - \sum_{i \in \mathbb{P}_n^{[j]}}\frac{(1 - A_i)\hat{\pi}^{[-j]}(X_i)}{(1 - \hat{\pi}^{[-j]}(X_i))\sum_{i \in \mathbb{P}_n^{[j]}}A_i}\left(Y_{i1} - \hat{\mu}_1^{[-j]}(X_i,0) - \frac{R_{i0}}{\hat{\gamma}^{[-j]}(X_i,0)}\left(Y_{i0} - \hat{\mu}_0^{[-j]}(X_i,0)\right)\right),
\end{aligned}
\tag{12}
$$

while under Assumption 2.1 and Assumption 2.4, it is equal to:

$$
\hat{\theta}^{[j]} = \sum_{i \in \mathbb{P}_n^{[j]}} \frac{A_i}{\sum_{i \in \mathbb{P}_n^{[j]}} A_i} \left( Y_{i1} - \left( \hat{\mu}_0^{[-j]}(X_i, Y_{i1}, 1) + \frac{R_{i0}}{\hat{\gamma}^{[-j]}(X_i, Y_{i1}, 1)} \left( Y_{i0} - \hat{\mu}_0^{[-j]}(X_i, Y_{i1}, 1) \right) \right) \right)
$$

$$
- \sum_{i \in \mathbb{P}_n^{[j]}} \frac{A_i}{\sum_{i \in \mathbb{P}_n^{[j]}} A_i} \left( \hat{\mu}_1^{[-j]}(X_i, 0) - \hat{\eta}_0^{[-j]}(X_i, 0) \right)
$$

$$
- \sum_{i \in \mathbb{P}_n^{[j]}} \frac{(1 - A_i)\hat{\pi}^{[-j]}(X_i)}{(1 - \hat{\pi}^{[-j]}(X_i)) \sum_{i \in \mathbb{P}_n^{[j]}} A_i} \left( Y_{i1} - \hat{\mu}_1^{[-j]}(X_i, 0) - \hat{\mu}_0^{[-j]}(X_i, Y_{i1}, 0) + \hat{\eta}_0^{[-j]}(X_i, 0) \right)
$$

$$
+ \sum_{i \in \mathbb{P}_n^{[j]}} \frac{(1 - A_i)\hat{\pi}^{[-j]}(X_i)}{(1 - \hat{\pi}^{[-j]}(X_i)) \sum_{i \in \mathbb{P}_n^{[j]}} A_i} \left( \frac{R_{i0}}{\hat{\gamma}^{[-j]}(X_i, Y_{i1}, 0)} \left( Y_{i0} - \hat{\mu}_0^{[-j]}(X_i, Y_{i1}, 0) \right) \right) .
$$

$$(13)$$

**Remark 3.1** (Multiple robustness). The structure of the estimators in Eq. 12 and Eq. 13 sheds light on their *multiple robustness* property. Under Assumption 2.3, we need either the model for $\mu^\star$ or both the models for $\pi^\star$ and $\gamma^\star$ to be well-specified in order to achieve consistency. Under Assumption 2.4, we also need the nested regression to be consistent if the propensity score is misspecified. See Appendix Table 1 for a description of the possible combinations of nuisance functions required for consistency.

### 3.1 ESTIMATING THE NESTED REGRESSION

The nested regression function $\eta_0^\star(x, 0) = \mathbb{E}\left[\mu_0^\star(x, Y_1, 0) \mid X = x, A = 0\right]$ that appears in our estimator under Assumption 2.4 is of particular interest. Here, we showcase a regression-based approach to estimate it, and we defer to Appendix Section B.2 a second approach based on conditional densities. For simplicity, we develop the arguments in this Section by assuming that the dataset is splitted in two folds, the former, $\hat{\mathbb{P}}$, being employed for nuisance training and the latter, $\mathbb{P}_n$, for influence function averaging.

In the regression-based approach, we treat $\mu_0^\star(x, Y_1, 0)$ as the response variable with $X$ and as predictor, and we fit a model to learn $\eta_0^\star(x, 0)$. Of course, $\mu_0^\star(x, Y_1, 0)$ is not known and thus has to be estimated on $\hat{\mathbb{P}}$. Interestingly, this nested approach shares many commonalities with the DR-Learner, a method commonly used to estimate heterogeneous treatment effects (Foster & Syrgkanis, 2023; Kennedy, 2023), and counterfactual regression (Yang et al., 2023b). First, we introduce some additional notation. Let $\hat{\eta}_0(x, 0) = \hat{\mathbb{E}}_n\left[\hat{\mu}_0(x, Y_1, 0) \mid X = x, A = 0\right]$ be the regression of $\hat{\mu}_0(x, Y_1, 0)$ on the covariates in the averaging sample $\mathbb{P}_n$, and $\tilde{\eta}_0(x, 0) = \hat{\mathbb{E}}_n\left[\mu_0^\star(x, Y_1, 0) \mid X = x, A = 0\right]$ be the corresponding oracle estimator regressing the true $\mu_0^\star(x, Y_1, 0)$ onto the covariates. Finally, denote the oracle risk $\tilde{\Delta}_n^2(x) = \mathbb{E}\left[(\tilde{\eta}_0(x, 0) - \eta_0^\star(x, 0))^2\right]$ and the conditional bias as $\hat{b}(x, y_1, 0) = \mathbb{E}\left[\hat{\mu}_0(X, Y_1, 0) - \mu_0^\star(X, Y_1, 0) \mid \hat{\mathbb{P}}, X = x, Y_1 = y_1\right]$. We can now introduce the definition of *stable estimator*.

**Definition 3.2** (Stability of estimator). *The regression estimator $\hat{\mathbb{E}}_n\left[\cdot \mid X = x, A = 0\right]$ is defined as stable at $X = x$ and $A = 0$ (with respect to a distance metric $d$) if*

$$
\frac{\hat{\eta}_0(X, 0) - \tilde{\eta}_0(X, 0) - \hat{\mathbb{E}}_n\left[\hat{b}(X, Y_1, 0) \mid X, A = 0\right]}{\tilde{\Delta}_n(X)} \xrightarrow{p} 0 , \tag{14}
$$

*as $d(\hat{\mu}_0(X, Y_1, 0), \mu_0^\star(X, Y_1, 0)) \xrightarrow{p} 0$.*

**Remark 3.3.** This pointwise (or local) definition, introduced by Kennedy (2023) and extended to $\mathbb{L}_2$-norm by Rambachan et al. (2022), is inspired by empirical processes and can be rewritten as

$$
\hat{\eta}_0(X, 0) - \tilde{\eta}_0(X, 0) = \hat{\mathbb{E}}_n\left[\hat{b}(X, Y_1, 0) \mid X, A = 0\right] + o_{\mathbb{P}^\star}\left(\tilde{\Delta}_n(X)\right) , \tag{15}
$$

as $d(\hat{\mu}_0(X, Y_1, 0), \mu_0^\star(X, Y_1, 0)) \xrightarrow{p} 0$.

When estimating the nested regression function $\eta_0^\star(x, 0) = \mathbb{E}\left[\mu_0^\star(x, Y_1, 0) \mid X = x, A = 0\right]$, a misspecified model for $\mu_0^\star(x, y_1, 0)$ would imply a misspecified model for $\eta_0^\star(x, 0)$. In fact, assuming that the regression estimator $\hat{\mathbb{E}}_n\left[\cdot \mid X = x, A = 0\right]$ is stable, the bias term shows first-order dependence on the estimation error in the nuisance function. To gain robustness, one can instead learn an estimate of the *augmented nested regression* $\mathbb{E}\left[\mu_0^\star(x, Y_1, 0) + R_0\left(Y_0 - \mu_0^\star(x, Y_1, 0)\right) / \gamma^\star(x, Y_1, 0) \mid X = x, A = 0\right]$. This strategy provides protection against misspecification in $\mu_0^\star(x, y_1, 0)$, provided that a good estimate $\gamma^\star(x, y_1, 0)$ is available. We can make this property more formal.

**Theorem 3.4** (Oracle property)**.** *Assume that the regression estimator $\hat{\mathbb{E}}_n\left[\cdot \mid X = x, A = 0\right]$ is stable with respect to distance $d$. Assume also that the estimated pseudo-outcomes converge in probability to truth, i.e.*

$$d(\hat{\mu}_0(x, Y_1, 0) + R_0\left(Y_0 - \hat{\mu}_0(x, Y_1, 0)\right) / \hat{\gamma}(x, Y_1, 0), \mu_0^\star(x, Y_1, 0) + R_0\left(Y_0 - \mu_0^\star(x, Y_1, 0)\right) / \gamma^\star(x, Y_1, 0)) \xrightarrow{p} 0\,.$$
(16)

*Then*

$$\hat{\eta}_0(X, 0) - \tilde{\eta}_0(X, 0) = \hat{\mathbb{E}}_n\left[\hat{b}(X, Y_1, 0) \mid X = x, A = 0\right] + o_{\mathbb{P}^\star}\left(\tilde{\Delta}_n(X)\right)\,,$$
(17)

*where*

$$\hat{b}(x, y_1, 0) = \frac{(\hat{\mu}_0(x, y_1, 0) - \mu_0^\star(x, y_1, 0))(\hat{\gamma}(x, y_1, 0) - \gamma^\star(x, y_1, 0))}{\hat{\gamma}(x, y_1, 0)}\,,$$
(18)

*and $\hat{\eta}_0(X, 0)$ is oracle efficient if $\hat{\mathbb{E}}_n\left[\hat{b}(X, Y_1, 0) \mid X = x, A = 0\right] = o_{\mathbb{P}^\star}\left(\tilde{\Delta}_n(x)\right)$.*

**Remark 3.5.** The previous Theorem gives conditions for achieving the oracle rate, which can be phrased in terms of nuisance smoothness and problem dimension. For example, if we assume that $\mu_0^\star(X, Y_1, 0)$ is $\alpha$-smooth in $X$ and $\beta$-smooth in $Y_1$, and $\gamma^\star(X, Y_1, 0)$ is $\rho$-smooth, then we can build nonparametric estimators $\hat{\mu}_0(X, Y_1, 0)$ and $\hat{\gamma}(X, Y_1, 0)$ whose convergence rates are $n^{-1/(2+p/\alpha+1/\beta)}$ and $n^{-1/(2+(p+1)/\rho)}$, respectively. By the previous Theorem, we can achieve the oracle rate whenever $\rho \geq (p+1)/[\alpha(2 + 1/\beta)(2 + p/\alpha + 1/\beta)/p - 2]$. This is relevant because, without augmentation, we would typically only achieve the oracle rate when $p/\alpha \to 0$, which holds only when $\alpha \to \infty$. In contrast, with augmentation, for each finite $\alpha$ we can find a suitable $\rho$ that satisfies the above oracle rate condition.

## 3.2 INFERENCE

We can now introduce a set of high-level assumptions that are useful for inferential purposes.

**Assumption 3.6** (Inference)**.** Let the number of cross-fitting folds be fixed at $J$, and assume that:

    **a.** There exist $\bar{\mu}, \bar{\pi}, \bar{\gamma}$, and $\bar{\eta}$ such that, for each $j \in \{1, \ldots, J\}$, one has

$$\varphi\left(\mathcal{D}; \theta^\star; \hat{\mu}^{[-j]}; \hat{\pi}^{[-j]}; \hat{\gamma}^{[-j]}; \hat{\eta}^{[-j]}\right) \xrightarrow{\mathbb{L}^2} \varphi\left(\mathcal{D}; \theta^\star; \bar{\mu}; \bar{\pi}; \bar{\gamma}; \bar{\eta}\right)\,.$$
(19)

    **b.** For each $j \in \{1, \ldots, J\}$, one has

$$\sqrt{n}\sum_{j=1}^{J} \kappa^{[j]} = o_{\mathbb{P}}(1)\,,$$
(20)

    where the *remainder term* $\kappa^{[j]}$ is defined as $\kappa^{[j]} = \theta\left(\hat{\mathbb{P}}^{[-j]}\right) - \theta(\mathbb{P}^\star) + \mathbb{E}\left[\varphi\left(\mathcal{D}; \theta^\star; \hat{\mu}^{[-j]}; \hat{\pi}^{[-j]}; \hat{\gamma}^{[-j]}; \hat{\eta}^{[-j]}\right)\right]$.

    **c.** Given $\xi > 0$, for each $j \in \{1, \ldots, J\}$, $\hat{\pi}^{[-j]}$ and $\hat{\gamma}^{[-j]}$ are bounded away from $\xi$ and $1 - \xi$ with probability 1.

**Remark 3.7.** These are all standard assumptions routinely employed in causal inference. See Kennedy (2024) for a review. Assumption **a** is used to control the empirical process term; Assumption **b** is used to control the remainder term; Assumption **c** is commonly referred to as *strong overlap*.

**Theorem 3.8** (Asymptotic normality and efficiency). *Under Assumption 3.6, the estimator $\hat{\theta}$ obtained by solving the estimating equation 10 is asymptotically normal and achieves the semiparametric efficiency bound as $n$ goes to infinity, that is*

$$\sqrt{n}\left(\hat{\theta} - \theta^{\star}\right) \rightsquigarrow \mathcal{N}\left(0, \mathbb{V}\left[\varphi\left(\mathcal{D}\right)\right]\right). \tag{21}$$

Endowed with a procedure for valid asymptotic inference, we now turn to the empirical validation of our estimators.

## 4 SIMULATION STUDY

To evaluate the finite-sample performance of our proposed estimators and to validate their theoretical properties, we conduct an extensive Monte Carlo simulation study. The primary goal is to assess the estimators' multiple robustness, particularly when some or all of the nuisance function models are misspecified. Our data generating process (DGP), which closely resembles the one pioneered by Kang & Schafer (2007) and then revisted by Sant'Anna & Zhao (2020), is designed to create scenarios where each model can be independently well or misspecified. For each simulation run, we generate a data set with a sample size of $n = 2000$ (Appendix Section C provides simulations for additional sample sizes). The true ATT is fixed at $\theta^{\star} = 5$. We repeat our simulations for 500 runs. The DGP is structured as follows:

- **Covariates.** We first generate a set of four true baseline covariates $Z = (Z_1, Z_2, Z_3, Z_4)$ from a standard normal distribution. From these, we create a corresponding set of four observed covariates $X = (X_1, X_2, X_3, X_4)$ by applying complex, non-linear transformations to $Z$ (e.g., involving exponential, polynomial, and interaction terms). This setup ensures that models based on $Z$ are correctly specified, while models using only $X$ are misspecified. In particular, the $X$'s are defined as

$$X_1 = \exp\left(\frac{Z_1}{2}\right), \quad X_2 = \frac{Z_2}{1 + \exp(Z_1)} + 10, \quad X_3 = \left(\frac{Z_1 Z_3}{25} + 0.6\right)^3, \quad X_4 = (Z_2 + Z_4 + 20)^2. \tag{22}$$

- **Treatment.** The treatment assignment indicator, $A$, is generated from a Bernoulli distribution with a probability (propensity score) that is a logistic function of the true covariates $Z$, that is

$$\text{logit}(\pi(Z)) = \text{logit}\left(\mathbb{P}\left[A = 1 \mid Z\right]\right) = -Z_1 + 0.5 Z_2 - 0.25 Z_3 - 0.1 Z_4. \tag{23}$$

- **Outcomes.** Pre-treatment outcomes are generated as linear functions of $Z$ plus standard normal error terms, that is

$$Y_0 = 210 + 27.4 Z_1 + 13.7 Z_2 + 13.7 Z_3 + 13.7 Z_4 + \varepsilon, \quad \text{where} \quad \varepsilon \sim \mathcal{N}(0, 1). \tag{24}$$

Post-treatment potential outcomes are then generated as

$$\begin{aligned} Y_1^{(0)} &= Y_0 + \varepsilon, \quad \text{where} \quad \varepsilon \sim \mathcal{N}(0, 1), \\ Y_1^{(1)} &= Y_1^{(0)} + A\theta^{\star}, \end{aligned} \tag{25}$$

which are then summarized, by consistency assumption, into the observed post-treatment outcome $Y_1 = A Y_1^{(1)} + (1 - A) Y_1^{(0)}$.

- **Missingness mechanism.** To align with our theoretical framework, we simulate two distinct missingness patterns for the pre-treatment outcome $Y_0$. Under Assumption 2.3, the missingness indicator $R_0$ is drawn from a Bernoulli distribution where the probability is a logistic function of the true covariates $Z$ and the treatment status $A$, i.e.

$$\text{logit}(\gamma(Z, A)) = \text{logit}\left(\mathbb{P}\left[R_0 = 1 \mid Z, A\right]\right) = -0.25 Z_1 - 0.1 Z_2 - 0.5 Z_3 + 0.3 Z_4 - 0.2 A. \tag{26}$$

Under Assumption 2.4, this probability additionally depends on the post-treatment outcome $Y_1$, creating a more complex logistic dependency:

$$\text{logit}(\gamma(Z, A, Y_1)) = \text{logit}\left(\mathbb{P}\left[R_0 = 1 \mid Z, A, Y_1\right]\right) = -0.25 Z_1 - 0.1 Z_2 - 0.5 Z_3 + 0.3 Z_4 - 0.2 A + 0.01 Y_1. \tag{27}$$

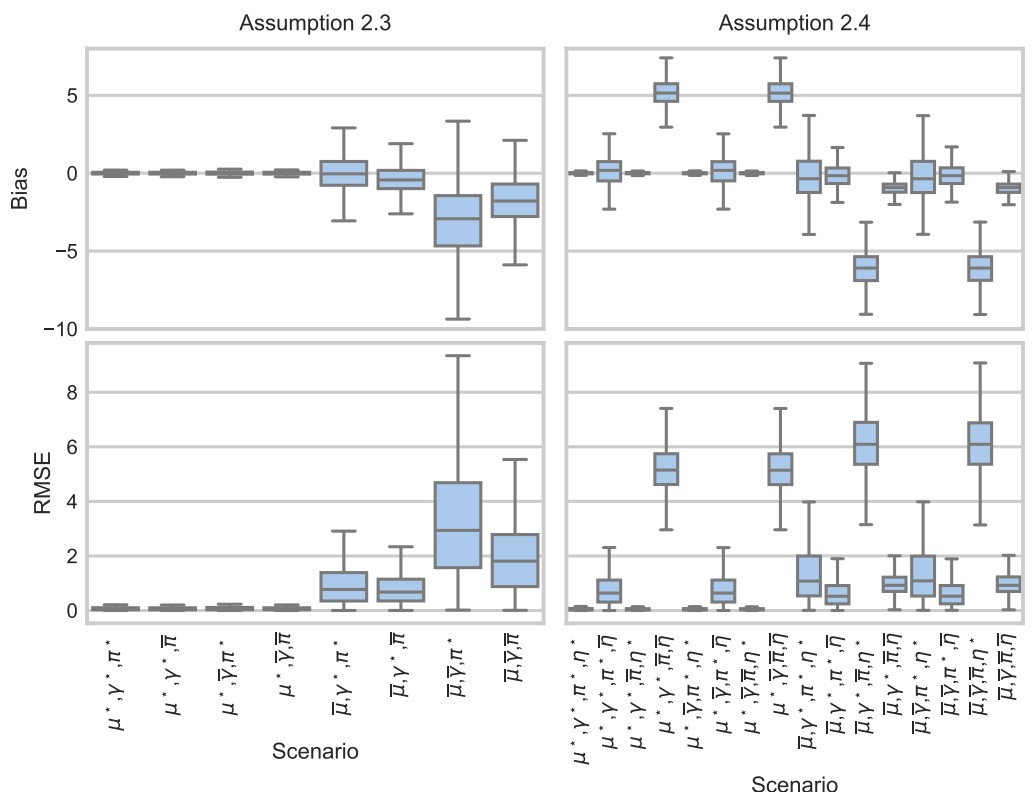

Figure 2: Simulation results. Boxplots of Bias (top row) and Root Mean Squared Error (RMSE) (bottom row) from 500 simulation runs. The left column shows the performance of the estimator under Assumption 2.3, while the right column shows the performance under Assumption 2.4. Each scenario on the x-axis represents a combination of correctly specified (indicated by a star $\star$) and misspecified nuisance functions (indicated by a bar). The results visually confirm the multiple robustness property of our estimators. Both bias and RMSE are negligible when a sufficient subset of nuisance models is correctly specified. Performance degrades significantly, as theory predicts, when the conditions for multiple robustness are violated.

For each of the 500 simulated datasets, we estimate the ATT using the influence function-based estimators in Equations 12 and 13. The nuisance functions are estimated using standard logistic and ordinary least squares models. To test the multiple robustness property, we cycle through all possible combinations of correctly specified and misspecified models for the nuisance functions. A model is correctly specified if it uses the true covariates $Z$ as predictors and misspecified if it uses the observed, non-linear covariates $X$.

We evaluate the performance of our estimators across these different specifications using two standard metrics: *bias*, which is the difference between estimated ATT $\hat{\theta}$ and true ATT $\theta^\star$; and *root mean squared error* (RMSE), which is the square root of the average squared difference between the estimate and the true value. RMSE penalizes large errors and captures both bias and variance, providing a comprehensive measure of estimator quality. In Appendix Section C, we also display simulation results for a third evaluation metric, *empirical coverage*.

The results, displayed in Figure 2 and Appendix Tables 2 and 3, provide strong evidence for the theoretical properties of our estimators. As predicted by theory, the bias and the RMSE are negligible for both estimators in all scenarios where the conditions for consistency are met. This holds true, for example, when the outcome model is correctly specified. Conversely, the estimators show a clear bias and higher RMSE in the theoretically inconsistent scenarios. This demonstrates the estimators' breaking point, which we further investigate with additional simulations in Appendix Section C.

Overall, the simulation results strongly support the validity and robustness of the proposed estimators.

## 5 CONCLUSIONS

The Difference-in-Differences (DiD) method is a cornerstone of applied research, yet its validity is often threatened by the practical challenge of missing outcome data – a problem that can introduce significant selection bias and invalidate standard estimators. This paper addresses this critical gap by developing a rigorous and comprehensive framework for DiD estimation when pre or post-treatment outcomes are missing at random (MAR). Drawing on semiparametric theory, we make several key contributions. First, we establish nonparametric identification of the Average Treatment Effect on the Treated (ATT) under two distinct and plausible MAR mechanisms: one where missingness is independent of the outcome conditional on covariates, and another where it may depend on the post-treatment outcome. For each setting, we derive the semiparametric efficiency bound, establishing a formal benchmark for asymptotic precision. We then propose novel estimators that achieve these bounds, ensuring asymptotic semiparametric efficiency. A critical feature of our estimators is their multiple robustness, which guarantees consistency as long as a subset of the nuisance function models is correctly specified, providing a layer of protection against model misspecification in practice.

The implications of this work are both theoretical and practical. Our framework provides applied researchers in economics, public health, and social sciences with a principled and efficient toolkit to conduct credible DiD analysis using incomplete panel data. By formally accounting for missing data, our estimators enhance the reliability of causal claims drawn from real-world observational studies where complete data is the exception rather than the rule.

While this paper focuses on the canonical two-group, two-period setting for clarity, the principles developed here open several avenues for future research. A natural next step is the extension of this framework to more complex scenarios, such as the staggered treatment adoption settings that have been the focus of much recent literature. Further investigation into the performance of different machine learning methods for the nuisance components, particularly the nested regression function, would also be valuable. Finally, incorporating our efficient estimators into standard DiD software would greatly facilitate their adoption by the broader research community, strengthening the quality and credibility of causal inference across disciplines.

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

# APPENDIX

## A    PROOF OF MAIN STATEMENTS

### A.1    PROOF OF LEMMA 2.7

*Proof.* Under Assumption 2.1 and Assumption 2.3, the following chain of equalities holds:

$$
\begin{aligned}
\theta^\star &= \mathbb{E}\left[Y_1^{(1)} - Y_1^{(0)} \mid A = 1\right] \\
&= \mathbb{E}\left[Y_1^{(1)} - Y_0 \mid A = 1\right] - \mathbb{E}\left[Y_1^{(0)} - Y_0 \mid A = 1\right] \\
&= \mathbb{E}\left[Y_1^{(1)} - \mathbb{E}\left[Y_0 \mid X, A = 1\right] \mid A = 1\right] - \mathbb{E}\left[\mathbb{E}\left[Y_1^{(0)} - Y_0 \mid X, A = 1\right] \mid A = 1\right] \\
&= \mathbb{E}\left[Y_1^{(1)} \mid A = 1\right] - \mathbb{E}\left[\mathbb{E}\left[Y_0 \mid X, A = 1\right] \mid A = 1\right] - \mathbb{E}\left[\mathbb{E}\left[Y_1^{(0)} - Y_0 \mid X, A = 0\right] \mid A = 1\right] \\
&= \mathbb{E}\left[Y_1 \mid A = 1\right] - \mathbb{E}\left[\mathbb{E}\left[Y_0 \mid X, A = 1, R_0 = 1\right] \mid A = 1\right] - \mathbb{E}\left[\mathbb{E}\left[Y_1 \mid X, A = 0\right] \mid A = 1\right] \\
&\quad + \mathbb{E}\left[\mathbb{E}\left[Y_0 \mid X, A = 0, R_0 = 1\right] \mid A = 1\right] \\
&= \mathbb{E}\left[Y_1 - \mathbb{E}\left[Y_0 \mid X, A = 1, R_0 = 1\right] - \mathbb{E}\left[Y_1 \mid X, A = 0\right] + \mathbb{E}\left[Y_0 \mid X, A = 0, R_0 = 1\right] \mid A = 1\right] \\
&= \mathbb{E}\left[\frac{A}{\mathbb{E}\left[A\right]}\left(Y_1 - \mathbb{E}\left[Y_0 \mid X, A = 1, R_0 = 1\right] - \mathbb{E}\left[Y_1 \mid X, A = 0\right] + \mathbb{E}\left[Y_0 \mid X, A = 0, R_0 = 1\right]\right)\right].
\end{aligned}
\tag{28}
$$

Under Assumption 2.1 and Assumption 2.4, the following chain of equalities holds:

$$
\begin{aligned}
\theta^\star &= \mathbb{E}\left[Y_1^{(1)} - Y_1^{(0)} \mid A = 1\right] \\
&= \mathbb{E}\left[Y_1^{(1)} - Y_0 \mid A = 1\right] - \mathbb{E}\left[Y_1^{(0)} - Y_0 \mid A = 1\right] \\
&= \mathbb{E}\left[Y_1^{(1)} - \mathbb{E}\left[Y_0 \mid X, Y_1, A = 1\right] \mid A = 1\right] - \mathbb{E}\left[\mathbb{E}\left[Y_1^{(0)} - Y_0 \mid X, A = 1\right] \mid A = 1\right] \\
&= \mathbb{E}\left[Y_1^{(1)} \mid A = 1\right] - \mathbb{E}\left[\mathbb{E}\left[Y_0 \mid X, Y_1, A = 1\right] \mid A = 1\right] - \mathbb{E}\left[\mathbb{E}\left[Y_1^{(0)} - Y_0 \mid X, A = 0\right] \mid A = 1\right] \\
&= \mathbb{E}\left[Y_1 \mid A = 1\right] - \mathbb{E}\left[\mathbb{E}\left[Y_0 \mid X, Y_1, A = 1, R_0 = 1\right] \mid A = 1\right] - \mathbb{E}\left[\mathbb{E}\left[Y_1 \mid X, A = 0\right] \mid A = 1\right] \\
&\quad + \mathbb{E}\left[\mathbb{E}\left[\mathbb{E}\left[Y_0 \mid X, Y_1, A = 0\right] \mid X, A = 0\right] \mid A = 1\right] \\
&= \mathbb{E}\left[Y_1 - \mathbb{E}\left[Y_0 \mid X, Y_1, A = 1, R_0 = 1\right] - \mathbb{E}\left[Y_1 \mid X, A = 0\right] + \mathbb{E}\left[\mathbb{E}\left[Y_0 \mid X, Y_1, A = 0, R_0 = 1\right] \mid X, A = 0\right] \mid A = 1\right] \\
&= \mathbb{E}\left[\frac{A}{\mathbb{E}[A]}\left(Y_1 - \mathbb{E}\left[Y_0 \mid X, Y_1, A = 1, R_0 = 1\right] - \mathbb{E}\left[Y_1 \mid X, A = 0\right] + \mathbb{E}\left[\mathbb{E}\left[Y_0 \mid X, Y_1, A = 0, R_0 = 1\right] \mid X, A = 0\right]\right)\right].
\end{aligned}
\tag{29}
$$

$\square$

### A.2 PROOF OF PROPOSITION 2.8

*Proof.* The semiparametric efficiency bound is given by the variance of the efficient influence function (Bickel et al., 1993). We therefore need to show that Equation 5 and Equation 6 are the efficient influence functions under Assumption 2.3 and Assumption 2.4, respectively. In a nonparametric model, the efficient influence function must satisfy the *Von Mises expansion*

$$
\hat{\theta}_{\text{plug-in}} - \theta^\star = -\mathbb{E}\left[\varphi\left(\mathcal{D}; \hat{\theta}_{\text{plug-in}}\right)\right] + \kappa_2(\hat{\mathbb{P}}, \mathbb{P}^\star),
\tag{30}
$$

where $\hat{\theta}_{\text{plug-in}}$ is a plug-in estimator of the expressions in Lemma 2.7 computed on a sample $\hat{\mathbb{P}}$, and $\kappa_2(\hat{\mathbb{P}}, \mathbb{P}^\star)$ is a *second-order remainder term* (which means it only depends on products or squares of differences between $\mathbb{P}^\star$ and $\hat{\mathbb{P}}$) – see Kennedy (2024) and Lemma 2 in Kennedy et al. (2023) for details. Therefore, we need to evaluate the remainder term $\kappa_2(\hat{\mathbb{P}}, \mathbb{P}^\star)$ and verify that it is second-order.

Under Assumption 2.3, the remainder term takes the form

$$
\begin{aligned}
\kappa_2(\hat{\mathbb{P}}, \mathbb{P}^\star) &= \hat{\theta}_{\text{plug-in}} - \theta^\star + \mathbb{E}\left[\varphi\left(\mathcal{D}; \hat{\theta}_{\text{plug-in}}\right)\right] \\
&= (\hat{\theta}_{\text{plug-in}} - \theta^\star)\left(1 - \frac{\mathbb{E}[A]}{\hat{\mathbb{E}}_n[A]}\right) \\
&\quad - \frac{1}{\hat{\mathbb{E}}_n[A]}\mathbb{E}\left[(\hat{\mu}_0(X,1) - \mu_0^\star(X,1))\left(1 - \frac{\gamma^\star(X,1)}{\hat{\gamma}(X,1)}\right)\right] \\
&\quad - \frac{1}{\hat{\mathbb{E}}_n[A]}\mathbb{E}\left[(\hat{\mu}_1(X,0) - \mu_1^\star(X,0))\left(\frac{\hat{\pi}(X) - \pi^\star(X)}{1 - \hat{\pi}(X)}\right)\right] \\
&\quad + \mathbb{E}\left[\frac{(1 - \pi^\star(X))\hat{\pi}(X)}{(1 - \hat{\pi}(X))\hat{\mathbb{E}}_n[A]}(\mu_0^\star(X,0) - \hat{\mu}_0(X,0))\left(\frac{\gamma^\star(X,0)}{\hat{\gamma}(X,0)} - 1\right)\right] \\
&\quad + \mathbb{E}\left[\frac{1}{\hat{\mathbb{E}}_n[A]}(\hat{\mu}_0(X,0) - \mu_0(X,0))\frac{\hat{\pi}(X) - \pi^\star(X)}{1 - \hat{\pi}(X)}\right].
\end{aligned}
\tag{31}
$$

Under Assumption 2.4, the remainder term takes the form

$$\kappa_2(\hat{\mathbb{P}}, \mathbb{P}^\star) = \hat{\theta}_{\text{plug-in}} - \theta^\star + \mathbb{E}\left[\varphi\left(\mathcal{D}; \hat{\theta}_{\text{plug-in}}\right)\right]$$

$$= (\hat{\theta}_{\text{plug-in}} - \theta^\star)\left(1 - \frac{\mathbb{E}[A]}{\hat{\mathbb{E}}_n[A]}\right)$$

$$- \frac{1}{\hat{\mathbb{E}}_n[A]}\mathbb{E}\left[(\hat{\mu}_0(X, Y_1, 1) - \mu_0^\star(X, Y_1, 1))\left(1 - \frac{\gamma^\star(X, Y_1, 1)}{\hat{\gamma}(X, Y_1, 1)}\right)\right]$$

$$- \frac{1}{\hat{\mathbb{E}}_n[A]}\mathbb{E}\left[(\hat{\mu}_1(X, 0) - \mu_1^\star(X, 0))\left(\frac{\hat{\pi}(X) - \pi^\star(X)}{1 - \hat{\pi}(X)}\right)\right] \quad (32)$$

$$+ \mathbb{E}\left[\frac{(1 - \pi^\star(X))\hat{\pi}(X)}{(1 - \hat{\pi}(X))\hat{\mathbb{E}}_n[A]}(\mu_0^\star(X, 0) - \hat{\mu}_0(X, 0))\left(\frac{\gamma^\star(X, Y_1, 0)}{\hat{\gamma}(X, Y_1, 0)} - 1\right)\right]$$

$$+ \mathbb{E}\left[\frac{1}{\hat{\mathbb{E}}_n[A]}(\hat{\eta}_0(X, 0) - \eta_0^\star(X, 0))\frac{\hat{\pi}(X) - \pi^\star(X)}{1 - \hat{\pi}(X)}\right].$$

Both remainders are second-order, and this completes the proof. $\qquad \square$

## A.3 PROOF OF THEOREM 3.4

*Proof.* By definition, stability and consistency together imply

$$\hat{\eta}_0(X, 0) - \tilde{\eta}_0(X, 0) = \hat{\mathbb{E}}_n\left[\hat{b}(X, Y_1, 0) \mid X, A = 0\right] + o_{\mathbb{P}^\star}\left(\tilde{\Delta}_n(X)\right), \quad (33)$$

where

$$\hat{b}(x, y_1, 0) = \frac{(\hat{\mu}_0(x, y_1, 0) - \mu_0^\star(x, y_1, 0))(\hat{\gamma}(x, y_1, 0) - \gamma^\star(x, y_1, 0))}{\hat{\gamma}(x, y_1, 0)}, \quad (34)$$

by iterated expectation. Therefore if $\hat{\mathbb{E}}_n\left[\hat{b}(X, Y_1, 0) \mid X = x, A = 0\right] = o_{\mathbb{P}^\star}\left(\tilde{\Delta}_n(x)\right)$ the result follows. $\qquad \square$

## A.4 PROOF OF THEOREM 3.8

For simplicity, we show the result for a single cross-fitting fold. In particular, we denote the distribution where the nuisance functions are trained as $\hat{\mathbb{P}}$ and the distribution where the influence functions are approximated as $\mathbb{P}_n$. Similarly, we denote as $\bar{\mu}$, $\bar{\pi}$, $\bar{\gamma}$, and $\bar{\eta}$ the population limits of $\hat{\mu}$, $\hat{\pi}$, $\hat{\gamma}$ and $\hat{\eta}$, respectively.

First, assume that $\mathbb{V}\left[\varphi\left(\mathcal{D}; \theta^\star; \bar{\mu}; \bar{\pi}; \bar{\gamma}; \bar{\eta}\right)\right] < \infty$ is known. By *Von Mises* expansion, we have

$$\hat{\theta} - \theta^\star = \mathbb{P}_n\left[\varphi\left(\mathcal{D}; \theta^\star; \bar{\mu}; \bar{\pi}; \bar{\gamma}; \bar{\eta}\right)\right]$$
$$+ (\mathbb{P}_n - \mathbb{P}^\star)\left[\varphi\left(\mathcal{D}; \theta^\star; \hat{\mu}; \hat{\pi}; \hat{\gamma}; \hat{\eta}\right) - \varphi\left(\mathcal{D}; \theta^\star; \bar{\mu}; \bar{\pi}; \bar{\gamma}; \bar{\eta}\right)\right] \quad (35)$$
$$+ \kappa_2(\hat{\mathbb{P}}, \mathbb{P}^\star).$$

We refer to the three elements on the right-hand side respectively as the *influence function term*, the *empirical process term* and the *remainder term*.

We analyze each component independently. The first influence function term on the right-hand side is a sum of mean 0, finite-variance random variables, with the right $n^{-1/2}$ scaling, and by the Central Limit Theorem this converges to a Normal distribution with mean 0 and variance equal to $\mathbb{V}\left[\varphi\left(\mathcal{D}; \theta^\star; \bar{\mu}; \bar{\pi}; \bar{\gamma}; \bar{\eta}\right)\right]$.

We now need to show that the CLT component dominates the other terms. In particular, we need to show that the second empirical process term is of order $O_{\mathbb{P}}\left(n^{-1/2}\right)$ and that the third remainder term is of order $O_{\mathbb{P}}\left(n^{-1/2}\right)$. The former follows from Lemma 3.7 in Testa et al. (2025) under Assumption 3.6, **a**; the latter directly follows from Assumption 3.6, **b**.

Summarizing the previous results, we have

$$\sqrt{n}\left(\hat{\theta} - \theta^{\star}\right) = \frac{1}{\sqrt{n}} \sum_{i=1}^{n} \varphi\left(\mathcal{D}_i; \theta^{\star}; \bar{\mu}; \bar{\pi}; \bar{\gamma}; \bar{\eta}\right) + o_{\mathbb{P}}(1), \tag{36}$$

from which the required CLT follows.

Finally, by Slutsky theorem, the previous result holds also when $\hat{\sigma}^2 \xrightarrow{p} \mathbb{V}\left[\varphi\left(\mathcal{D}; \theta^{\star}; \bar{\mu}; \bar{\pi}; \bar{\gamma}; \bar{\eta}\right)\right]$ replaces $\mathbb{V}\left[\varphi\left(\mathcal{D}; \theta^{\star}; \bar{\mu}; \bar{\pi}; \bar{\gamma}; \bar{\eta}\right)\right]$.

# B ADDITIONAL THEORETICAL AND PRACTICAL REMARKS

| Assumption 2.3 | Assumption 2.4 |
|---|---|
| $\mu_0^{\star}(X,1)$ or $\gamma^{\star}(X,1)$ | $\mu_0^{\star}(X,Y_1,1)$ or $\gamma^{\star}(X,Y_1,1)$ |
| $\mu_1^{\star}(X,0)$ or $\pi^{\star}(X)$ | $\mu_1^{\star}(X,0)$ or $\pi^{\star}(X)$ |
| $\mu_0^{\star}(X,0)$ or $\gamma^{\star}(X,0)$ | $\mu_0^{\star}(X,Y_1,0)$ or $\gamma^{\star}(X,Y_1,0)$ |
| $\mu_0^{\star}(X,0)$ or $\pi^{\star}(X)$ | $\eta_0^{\star}(X,0)$ or $\pi^{\star}(X)$ |

Table 1: Summary of multiple robustness property

## B.1 PRACTICAL ESTIMATION OF THE NESTED REGRESSION $\eta_0^*$

The estimation of the nested regression function, $\eta_0^{\star}(x,0) = \mathbb{E}\left[\mu_0^{\star}(x,Y_1,0) \mid X = x, A = 0\right]$, is a critical component for the estimator under Assumption 2.4. As discussed in Section 3.1, this is a non-trivial, "DR-Learner" style problem (Kennedy, 2023). We provide two key pieces of practical guidance: an algorithm for the robust augmented estimator (Theorem 3.4) and recommendations for implementation.

### B.1.1 ALGORITHM FOR AUGMENTED NESTED REGRESSION

To protect against misspecification of $\hat{\mu}_0$, we recommend estimating $\eta_0^{\star}$ using the augmented pseudo-outcome strategy from Theorem 3.4. This approach uses both $\hat{\mu}_0$ and $\hat{\gamma}$ to create a robust target for the final regression. Algorithm 1 provides a concrete, cross-fitted strategy.

### B.1.2 IMPLEMENTATION GUIDANCE

Our semiparametric framework, combined with cross-fitting, is specifically designed to accommodate flexible, data-adaptive machine learning (ML) models for the nuisance functions. For regression tasks (e.g., $\hat{\mu}_0$, $\hat{\mu}_1$, and the final $\hat{\eta}_0$), models like Random Forests or Gradient-Boosted Trees (GBTs) are excellent default choices. For propensity scores (e.g., $\hat{\pi}$, $\hat{\gamma}$), ML classifiers (e.g., Random Forest Classifier, Logistic Regression with flexible features) are recommended. In our own real-data application in Appendix Section E, we use Random Forests for all nuisance functions, which demonstrates their practical feasibility and effectiveness.

The estimators involve inverse probability weighting, particularly in the augmentation step. These terms can become unstable if the estimated probability $\hat{\gamma}$ is near zero. In such cases, it is standard practice to clip (or trim) predicted probabilities to prevent extreme weights. We recommend clipping all estimated probabilities (e.g., $\hat{\pi}$, $\hat{\gamma}$) to be bounded within a small range, for example, $[\epsilon, 1 - \epsilon]$ where $\epsilon = 0.01$ or $\epsilon = 0.05$. This small amount of trimming ensures numerical stability at the cost of introducing a negligible, finite-sample bias, a standard trade-off in robust estimation.

## B.2 CONDITIONAL DENSITY APPROACH

A second approach is to write the nested regression as

$$\eta_0^{\star}(x,0) = \int \mu_0^{\star}(x,y_1,0)p(y_1|X = x, A = 0)\, dy_1, \tag{37}$$

---

**Algorithm 1** Cross-Fitted Augmented Estimator for $\hat{\eta}_0(x, 0)$

---

1: **Input:** Full dataset $\mathcal{D} = \{\mathcal{D}_i\}_{i=1}^n$, Number of folds $J$ (e.g., $J = 5$ or 10).
2: Partition the $n$ observations into $J$ disjoint folds: $\mathcal{F}_1, \ldots, \mathcal{F}_J$.
3: Initialize an empty list $\mathcal{D}_{\text{pseudo}}$ to store pseudo-outcomes.
4: **for** $j = 1$ **to** $J$ **do**
5:     Define the training sample $\mathcal{D}^{[-j]} = \mathcal{D} \setminus \mathcal{F}_j$ (all data except fold $j$).
6:     Define the estimation sample $\mathcal{D}^{[j]} = \mathcal{F}_j$ (only data in fold $j$).
7:     {Train nuisance models on data **not** in fold $j$}
8:     Train $\hat{\mu}_0^{[-j]}(X, Y_1, 0)$ using $\mathcal{D}^{[-j]}$.
9:     Train $\hat{\gamma}^{[-j]}(X, Y_1, 0)$ using $\mathcal{D}^{[-j]}$.
10:    {Generate pseudo-outcomes for observations **in** fold $j$}
11:    **for** each observation $i \in \mathcal{D}^{[j]}$ **do**
12:       Predict $\hat{\mu}_i = \hat{\mu}_0^{[-j]}(X_i, Y_{i1})$.
13:       Predict $\hat{\gamma}_i = \hat{\gamma}^{[-j]}(X_i, Y_{i1})$.
14:       {Clip $\hat{\gamma}_i$ for stability (see Section B.1.2)}
15:       $\hat{\gamma}_i \leftarrow \max(\epsilon, \hat{\gamma}_i)$ and $\hat{\gamma}_i \leftarrow \min(1 - \epsilon, \hat{\gamma}_i)$ for a small $\epsilon > 0$.
16:       {Compute the augmented pseudo-outcome from Theorem 3.4}
17:       $\tilde{Y}_{i,\text{aug}} \leftarrow \hat{\mu}_i + \frac{R_{i0}}{\hat{\gamma}_i}(Y_{i0} - \hat{\mu}_i)$
18:       {Store the result}
19:       Add $(X_i, \tilde{Y}_{i,\text{aug}})$ to $\mathcal{D}_{\text{pseudo}}$.
20:    **end for**
21: **end for**
22: {Train the final nested regression model}
23: Train $\hat{\eta}_0(x, 0)$ by regressing the generated pseudo-outcomes $\tilde{Y}_{i,\text{aug}}$ on their corresponding covariates $X_i$ using the full dataset $\mathcal{D}_{\text{pseudo}}$.
24: **Output:** The final estimator $\hat{\eta}_0(x, 0)$.

---

and approximate the integral by first estimating the conditional density $p(y_1|X = x, A = 0)$ and averaging over the values of $y_1$. Several methods have been proposed to achieve this, both from classical nonparametric statistics literature, such as histograms and kernels (Wasserman, 2006), and from more recent advancements, such as FlexCode, which rolls back the conditional density estimation problem to a series of regression problems (Izbicki & Lee, 2017). With FlexCode, one first picks a system of orthonormal basis $\{\phi_j\}_{j=1}^\infty$, and then writes

$$p(y_1|X = x, A = 0) = \sum_{j=1}^\infty \beta_j(x, 0)\phi_j(y_1), \tag{38}$$

where one can show that the basis expansion coefficient appearing in the previous equation is equal to $\beta_j(x, 0) = \mathbb{E}\left[\phi_j(Y_1) \mid X = x, A = 0\right]$. Therefore, one can estimate these regressions individually and sum them back to get an estimate of the conditional distribution.

## C   ADDITIONAL SIMULATION RESULTS

### C.1   SIGNAL-TO-NOISE SIMULATION STUDY

To further validate our theoretical claims, we conduct an additional simulation study designed to analyze the empirical coverage and confidence interval (CI) width of our estimator, and visualize the precise "breaking points" of our method when the multiple robustness conditions fail. In particular, we move beyond the "all correct vs. all incorrect" simulation from the main text, measuring the performance of our estimator (under Assumption 2.3 for simplicity) in a setting where the nuisance functions are misspecified to varying controlled degrees. This approach has proved successful and insightful in other simulation studies, such as Das et al. (2024); Testa et al. (2025); Xu et al. (2025).

Table 2: Simulation results for Assumption 2.3, $n = 2000$.

| Nuisance Model Specification | | | Bias | RMSE |
|:---:|:---:|:---:|:---:|:---:|
| $\mu^\star$ correct | $\pi^\star$ correct | $\gamma^\star$ correct | | |
| ✓ | ✓ | ✓ | 0.004 | 0.103 |
| ✓ | ✗ | ✓ | 0.002 | 0.099 |
| ✓ | ✓ | ✗ | 0.007 | 0.115 |
| ✓ | ✗ | ✗ | 0.003 | 0.104 |
| ✗ | ✓ | ✓ | 0.053 | 1.363 |
| ✗ | ✗ | ✓ | 0.280 | 1.095 |
| ✗ | ✓ | ✗ | 3.328 | 4.082 |
| ✗ | ✗ | ✗ | 1.739 | 2.312 |

Table 3: Simulation results for Assumption 2.4, $n = 2000$.

| Nuisance Model Specification | | | | Bias | RMSE |
|:---:|:---:|:---:|:---:|:---:|:---:|
| $\mu^\star$ correct | $\gamma^\star$ correct | $\pi^\star$ correct | $\eta^\star$ correct | | |
| ✓ | ✓ | ✓ | ✓ | 0.000 | 0.078 |
| ✓ | ✓ | ✓ | ✗ | 0.020 | 1.132 |
| ✓ | ✓ | ✗ | ✓ | 0.000 | 0.074 |
| ✓ | ✓ | ✗ | ✗ | 5.197 | 5.197 |
| ✓ | ✗ | ✓ | ✓ | 0.000 | 0.078 |
| ✓ | ✗ | ✓ | ✗ | 0.021 | 1.133 |
| ✓ | ✗ | ✗ | ✓ | 0.001 | 0.073 |
| ✓ | ✗ | ✗ | ✗ | 5.197 | 5.197 |
| ✗ | ✓ | ✓ | ✓ | 0.156 | 1.869 |
| ✗ | ✓ | ✓ | ✗ | 0.136 | 0.830 |
| ✗ | ✓ | ✗ | ✓ | 6.176 | 6.176 |
| ✗ | ✓ | ✗ | ✗ | 0.980 | 0.996 |
| ✗ | ✗ | ✓ | ✓ | 0.155 | 1.866 |
| ✗ | ✗ | ✓ | ✗ | 0.135 | 0.831 |
| ✗ | ✗ | ✗ | ✓ | 6.178 | 6.178 |
| ✗ | ✗ | ✗ | ✗ | 0.979 | 0.996 |

### C.1.1 SETUP

We first describe the data generating process. We set the true ATT at $\theta^\star = 5$. Instead of generating covariates and combining them into nuisance functions, we directly sample the nuisance components and we treat them as ground truths. This allows us to precisely control the degree of nuisance model misspecification. We therefore the dependence on $X$. A baseline mean $\mu_0^\star(0)$ is drawn from $\mathcal{N}(5, 1)$, and a baseline effect $b$ is drawn from $\mathcal{N}(2, 0.5)$. The potential outcome regression functions are set as $\mu_0^\star(0) = \mu_0^\star(1)$, $\mu_1^\star(0) = \mu_0^\star(0) + b$, and $\mu_1^\star(1) = \mu_1^\star(0) + \theta^\star$. The true treatment probability $\pi^\star$ and missingness probabilities $\gamma^\star(0)$ and $\gamma^\star(1)$ are drawn from $\mathcal{U}(0.1, 0.9)$. Using these true nuisance functions, a dataset of $n = 1000$ observations is generated. The treatment indicator $A$ is drawn from a Bernoulli($\pi^\star$). Potential outcomes $\tilde{Y}_0$, $Y_{11}$, and $Y_{10}$ are generated by adding standard normal noise $\mathcal{N}(0, 1)$ to their respective true means. The observed post-treatment outcome $Y_1$ is set to $Y_{11}$ if $A = 1$ and $Y_{10}$ if $A = 0$. The missingness indicator $R_0$ is drawn from Bernoulli($\gamma^\star(1)$) if $A = 1$ and Bernoulli($\gamma^\star(0)$) if $A = 0$. The observed pre-treatment outcome $Y_0$ is set to $\tilde{Y}_0$ if $R_0 = 1$ and NA otherwise.

We then define three *correctness* parameters, $\alpha_\mu, \alpha_\pi, \alpha_\gamma \in [0, 1]$, which control the quality of the nuisance function estimates by defining estimates as convex combinations between true population quantities and random noise:

$$\hat{\mu}_t(a) = \alpha_\mu \mu_t^\star(a) + (1 - \alpha_\mu)\varepsilon_\mu, \quad \hat{\pi} = \alpha_\pi \pi^\star + (1 - \alpha_\pi)\varepsilon_\pi, \quad \hat{\gamma}(a) = \alpha_\gamma \gamma^\star(a) + (1 - \alpha_\gamma)\varepsilon_\gamma, \quad (39)$$

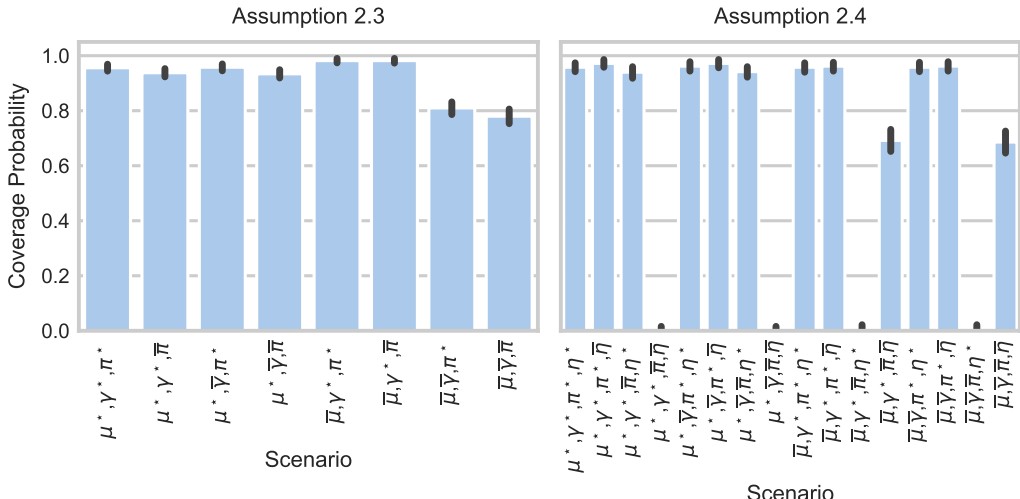

Figure 3: Simulation results with $n = 2000$. Barplot of empirical coverage from 500 simulation runs. The left column shows the performance of the estimator under Assumption 2.3, while the right column shows the performance under Assumption 2.4. Each scenario on the x-axis represents a combination of correctly specified (indicated by a star $\star$) and misspecified nuisance functions (indicated by a bar). The results visually confirm the multiple robustness property of our estimators.

where $t, a \in \{0, 1\}$, $\varepsilon_\mu$, $\varepsilon_\pi$, and $\varepsilon_\gamma$ are appropriate noise components, i.e. $\varepsilon_\mu \sim \mathcal{N}(0, 1)$, $\varepsilon_\pi, \varepsilon_\gamma \sim \mathcal{U}(0.1, 0.9)$. An $\alpha$ value of 1 means the model is perfectly specified (equal to the truth, $\mu_t^\star(a), \pi^\star, \gamma^\star(a)$), while an $\alpha$ value of 0 means the model is pure noise. The $\alpha$ parameters can be interpreted as rates of convergence of the nuisance functions to the corresponding population quantities. We repeat this experiment for 1000 simulations for each combination of $\alpha$ parameters on a grid from 0 to 1.

For each simulation, we compute the ATT estimate and a 95% confidence interval using Gaussian approximation. We then report four metrics averaged over the 1000 simulations: absolute bias, root mean squared error (RMSE), empirical coverage (the fraction of CIs containing the true ATT), and CI width.

### C.1.2  RESULTS

The results are presented in Figure 6. There, rows correspond to the degree of correctness of the propensity score model ($\alpha_\pi$). The four columns show the four metrics. Within each heatmap, the y-axis shows the degree of correctness of the outcome models ($\alpha_\mu$), and the x-axis shows the degree of correctness of the missingness models ($\alpha_\gamma$).

Our analysis of these results confirms the strong theoretical properties of our estimator. The first two columns (bias and RMSE) visually confirm the multiple robustness property. As predicted by theory for our estimator under Assumption 2.4, the estimator is consistent (bias and RMSE are near-zero) if:

- *Either* the outcome models are correct ($\alpha_\mu = 1$), which corresponds to the bottom row of each heatmaps. Note that bias is low regardless of the values of $\alpha_\pi$ or $\alpha_\gamma$.

- *Or* the propensity score *and* missingness models are correct ($\alpha_\pi = 1$ and $\alpha_\gamma = 1$). This corresponds to the last column of the heatmaps in the last row ($\alpha_\pi = 1$). The estimator is consistent here even when the outcome model is pure noise ($\alpha_\mu = 0$).

Conversely, when this condition fails – for instance, in the top-left heatmap where all models are partially or fully misspecified (e.g., $\alpha_\mu = 0.0, \alpha_\gamma = 0.0, \alpha_\pi = 0.0$) – the bias is massive (13.0), demonstrating the estimator's breaking point.

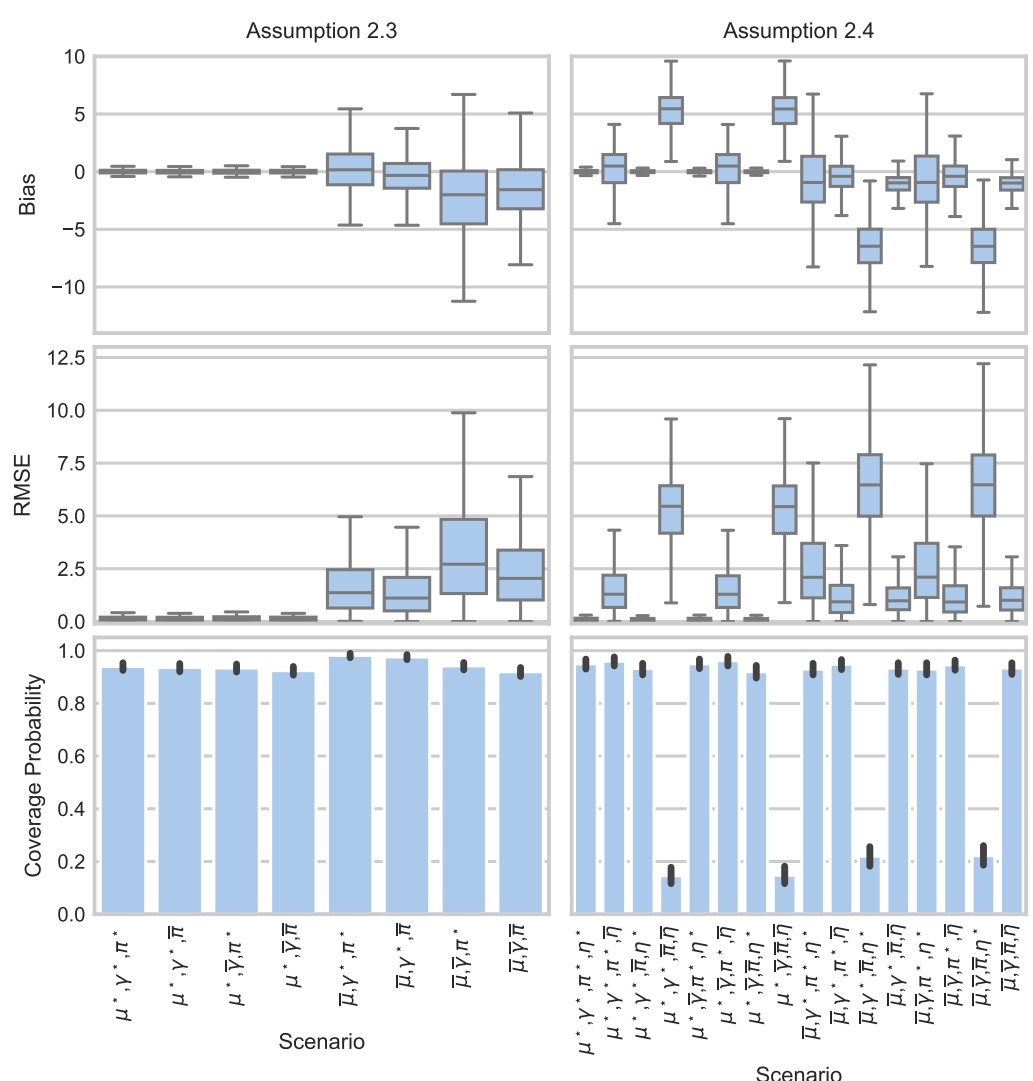

Figure 4: Simulation results with $n = 500$. Boxplots of bias (top row), root mean squared error (RMSE) (medium row), and barplot of empirical coverage (bottom row) from 500 simulation runs. The left column shows the performance of the estimator under Assumption 2.3, while the right column shows the performance under Assumption 2.4. Each scenario on the x-axis represents a combination of correctly specified (indicated by a star ⋆) and misspecified nuisance functions (indicated by a bar). The results visually confirm the multiple robustness property of our estimators. Both bias and RMSE are negligible when a sufficient subset of nuisance models is correctly specified. Performance degrades significantly, as theory predicts, when the conditions for multiple robustness are violated.

The third column on empirical coverage measures the inference performance of our estimator. When sufficient nuisance models are well-specified (i.e., when multiple robustness holds), empirical coverage is nominal. This is visible in the bottom row of all heatmaps ($\alpha_\mu = 1$) and in the last column of the heatmap in the last row ($\alpha_\pi = 1$, $\alpha_\gamma = 1.0$). This confirms that the Gaussian approximation is reliable and provides valid inference when the estimator is consistent at sufficiently fast product rates.

Furthermore, the heatmaps precisely show how the confidence intervals mis-cover when the robustness conditions fail. For instance, in the heatmap in the first row ($\alpha_\pi = 0$), if $\hat{\mu}$ is misspecified (e.g., $\alpha_\mu = 0$), the coverage collapses to the range 0.11–0.35, depending on the quality of $\hat{\gamma}$. Even if $\hat{\pi}$ is

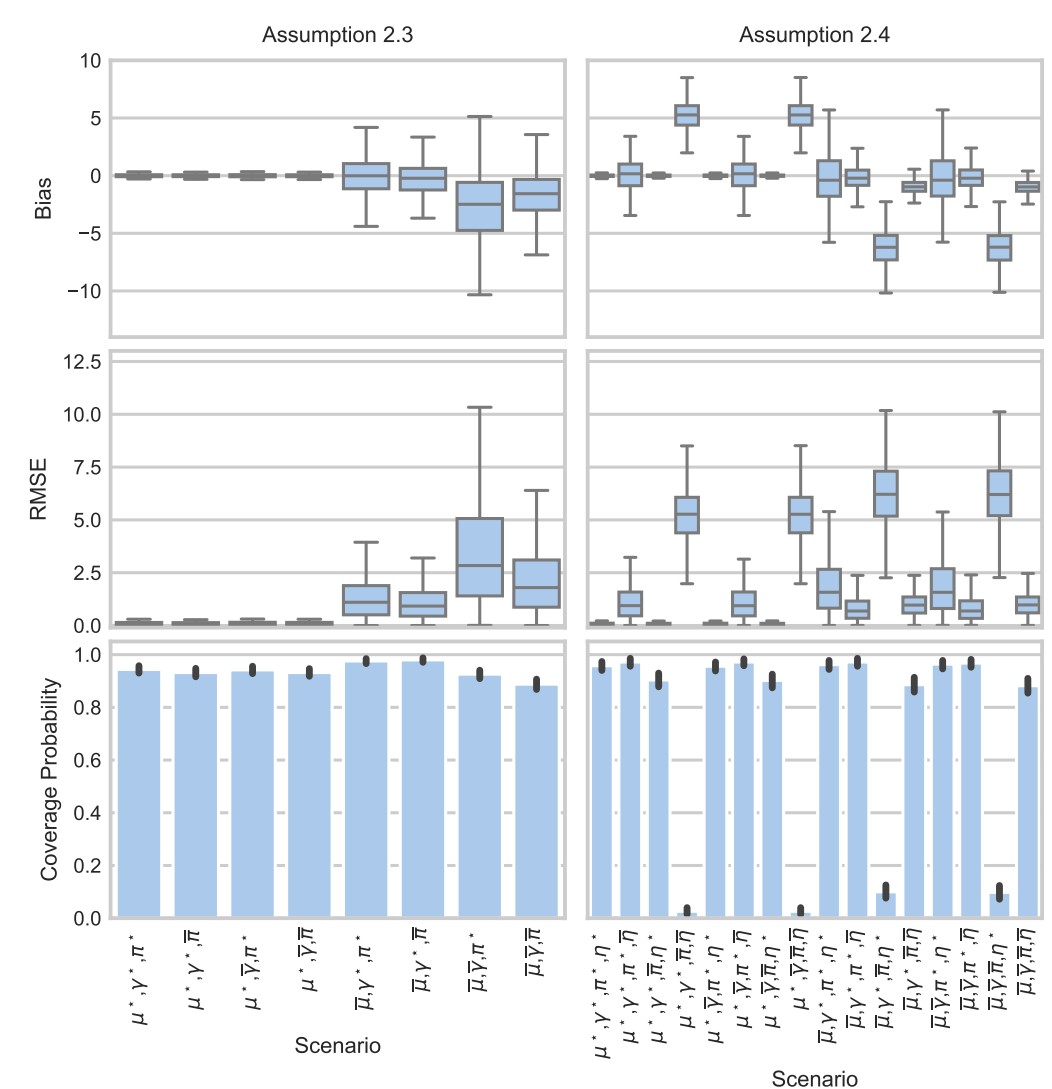

Figure 5: Simulation results with $n = 1000$. Boxplots of bias (top row), root mean squared error (RMSE) (medium row), and barplot of empirical coverage (bottom row) from 500 simulation runs. The left column shows the performance of the estimator under Assumption 2.3, while the right column shows the performance under Assumption 2.4. Each scenario on the x-axis represents a combination of correctly specified (indicated by a star ⋆) and misspecified nuisance functions (indicated by a bar). The results visually confirm the multiple robustness property of our estimators. Both bias and RMSE are negligible when a sufficient subset of nuisance models is correctly specified. Performance degrades significantly, as theory predicts, when the conditions for multiple robustness are violated.

correct ($\alpha_\pi = 1$, bottom heatmap), if $\hat{\mu}$ and $\hat{\gamma}$ are both incorrect, the coverage still collapses (down to 0.19). This demonstrates that for inference to be valid, the multiple robustness condition must be met.

Finally, the fourth column, CI width, measures the efficiency of our estimator. The CI is narrowest (most efficient, width approx. 0.9) when all models are correctly specified (bottom-right cell of the last heatmap: $\alpha_\mu = 1, \alpha_\pi = 1, \alpha_\gamma = 1$). The CI is wider (less efficient, width approx. 1.2–1.5) when relying on only the outcome model for consistency (bottom row of the top-right heatmap). This result empirically confirms that while our estimator provides valid inference (correct coverage) as

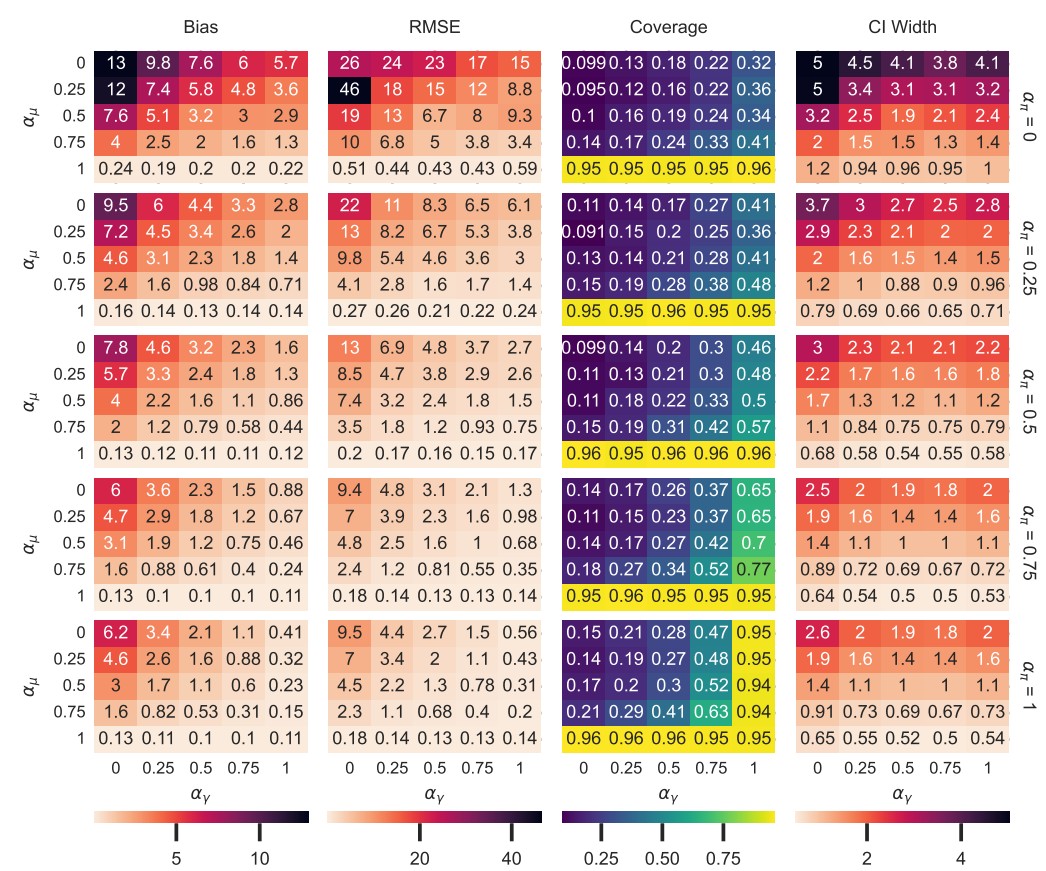

Figure 6: Performance of the ATT estimator under varying degrees of nuisance function misspecification. The five rows correspond to the correctness of the propensity score model ($\alpha_\pi$). The columns show bias, RMSE, empirical coverage, and CI width. Within each heatmap, the y-axis shows the correctness of the outcome model ($\alpha_\mu$) and the x-axis shows the correctness of the missingness model ($\alpha_\gamma$). This visualization confirms the estimator's multiple robustness property and pinpoints its breaking points.

long as the multiple robustness condition holds, there are significant efficiency gains from correctly specifying all nuisance functions.

## D    PARTIALLY MISSING POST-TREATMENT OUTCOME

### D.1    WHEN PRE-TREATMENT OUTCOME IS ALWAYS OBSERVED

**Assumption D.1** (Outcome independent missing at random). Let the following MAR assumptions hold:

   **a. No unmeasured confounding.** $Y_1 \perp\!\!\!\perp R_1 \mid X, A$.

   **b. Weak overlap.** $\gamma^\star(x, a) = \mathbb{P}[R_1 = 1 \mid X = x, A = a] \in (0, 1)$ almost surely for every $x \in \mathbb{R}^p$ and $a \in \{0, 1\}$.

**Assumption D.2** (Outcome dependent missing at random). Let the following MAR assumptions hold:

   **a. No unmeasured confounding.** $Y_1 \perp\!\!\!\perp R_1 \mid X, A, Y_0$.

**b. Weak overlap.** $\gamma^\star(x, y_0, a) = \mathbb{P}[R_1 = 1 \mid X = x, Y_0 = y_0, A = a] \in (0,1)$ almost surely for every $x \in \mathbb{R}^p$, $y_0 \in \mathbb{R}$, and $a \in \{0, 1\}$.

Equipped with the previous sets of assumptions, we can now identify the ATT as a function of the observed data as shown in the following Lemma.

**Lemma D.3** (Identification of ATT). *Under Assumption 2.1 and Assumption D.1, the ATT can be identified as a function of the observed data:*

$$\theta^\star = \mathbb{E}\left[\frac{A}{\mathbb{E}[A]}\left(\mathbb{E}[Y_1 \mid X, A = 1, R_1 = 1] - Y_0\right)\right]$$
$$- \mathbb{E}\left[\frac{A}{\mathbb{E}[A]}\left(\mathbb{E}[Y_1 \mid X, A = 0, R_1 = 1] - \mathbb{E}[Y_0 \mid X, A = 0]\right)\right]. \tag{40}$$

*Under Assumption 2.1 and Assumption D.2, the ATT can be identified as a function of the observed data:*

$$\theta^\star = \mathbb{E}\left[\frac{A}{\mathbb{E}[A]}\left(\mathbb{E}[Y_1 \mid X, Y_0, A = 1, R_1 = 1] - Y_0\right)\right]$$
$$- \mathbb{E}\left[\frac{A}{\mathbb{E}[A]}\left(\mathbb{E}[\mathbb{E}[Y_1 \mid X, Y_0, A = 0, R_1 = 1] \mid X, A = 0] - \mathbb{E}[Y_0 \mid X, A = 0]\right)\right]. \tag{41}$$

*Proof.* Under Assumption 2.1 and Assumption D.1, the following chain of equalities holds:

$$\theta^\star = \mathbb{E}\left[Y_1^{(1)} - Y_1^{(0)} \mid A = 1\right]$$

$$= \mathbb{E}\left[Y_1^{(1)} - Y_0 \mid A = 1\right] - \mathbb{E}\left[Y_1^{(0)} - Y_0 \mid A = 1\right]$$

$$= \mathbb{E}\left[\mathbb{E}\left[Y_1^{(1)} \mid X, A = 1\right] - Y_0 \mid A = 1\right] - \mathbb{E}\left[\mathbb{E}\left[Y_1^{(0)} - Y_0 \mid X, A = 1\right] \mid A = 1\right]$$

$$= \mathbb{E}[\mathbb{E}[Y_1 \mid X, A = 1, R_1 = 1] \mid A = 1] - \mathbb{E}[Y_0 \mid A = 1] - \mathbb{E}\left[\mathbb{E}\left[Y_1^{(0)} - Y_0 \mid X, A = 0\right] \mid A = 1\right]$$

$$= \mathbb{E}[\mathbb{E}[Y_1 \mid X, A = 1, R_1 = 1] \mid A = 1] - \mathbb{E}[Y_0 \mid A = 1] - \mathbb{E}[\mathbb{E}[Y_1 \mid X, A = 0, R_1 = 1] \mid A = 1]$$
$$+ \mathbb{E}[\mathbb{E}[Y_0 \mid X, A = 0] \mid A = 1]$$

$$= \mathbb{E}\left[\frac{A}{\mathbb{E}[A]}\left(\mathbb{E}[Y_1 \mid X, A = 1, R_1 = 1] - Y_0 - \mathbb{E}[Y_1 \mid X, A = 0, R_1 = 1] + \mathbb{E}[Y_0 \mid X, A = 0]\right)\right]. \tag{42}$$

Under Assumption 2.1 and Assumption D.2, the following chain of equalities holds:

$$\theta^\star = \mathbb{E}\left[Y_1^{(1)} - Y_1^{(0)} \mid A = 1\right]$$

$$= \mathbb{E}\left[Y_1^{(1)} - Y_0 \mid A = 1\right] - \mathbb{E}\left[Y_1^{(0)} - Y_0 \mid A = 1\right]$$

$$= \mathbb{E}\left[\mathbb{E}\left[Y_1^{(1)} \mid X, Y_0, A = 1\right] - Y_0 \mid A = 1\right] - \mathbb{E}\left[\mathbb{E}\left[Y_1^{(0)} - Y_0 \mid X, A = 1\right] \mid A = 1\right]$$

$$= \mathbb{E}\left[\mathbb{E}\left[Y_1^{(1)} \mid X, Y_0, A = 1\right] \mid A = 1\right] - \mathbb{E}[Y_0 \mid A = 1] - \mathbb{E}\left[\mathbb{E}\left[Y_1^{(0)} - Y_0 \mid X, A = 0\right] \mid A = 1\right]$$

$$= \mathbb{E}\left[\mathbb{E}\left[Y_1^{(1)} \mid X, Y_0, A = 1, R_1 = 1\right] \mid A = 1\right] - \mathbb{E}[Y_0 \mid A = 1] - \mathbb{E}[\mathbb{E}[\mathbb{E}[Y_1 \mid X, Y_0, A = 0] \mid X, A = 0] \mid A = 1]$$
$$+ \mathbb{E}[\mathbb{E}[Y_0 \mid X, A = 0] \mid A = 1]$$

$$= \mathbb{E}[\mathbb{E}[Y_1 \mid X, Y_0, A = 1, R_1 = 1] - Y_0 - \mathbb{E}[\mathbb{E}[Y_1 \mid X, Y_0, A = 0, R_1 = 1] \mid X, A = 0] + \mathbb{E}[Y_0 \mid X, A = 0] \mid A = 1]$$

$$= \mathbb{E}\left[\frac{A}{\mathbb{E}[A]}\left(\mathbb{E}[Y_1 \mid X, Y_0, A = 1, R_1 = 1] - Y_0 - \mathbb{E}[\mathbb{E}[Y_1 \mid X, Y_0, A = 0, R_1 = 1] \mid X, A = 0] + \mathbb{E}[Y_0 \mid X, A = 0]\right)\right]. \tag{43}$$

$\square$

We can also derive the efficient influence functions for the previous identified targets. Their variances define the semiparametric efficiency bounds.

**Proposition D.4** (Semiparametric efficiency bounds). *Under Assumption 2.1 and Assumption D.1, the efficient observed-data influence function is given by:*

$$\varphi\left(\mathcal{D}\right) = \frac{A}{\mathbb{E}\left[A\right]}\left(\left(\mu_1^\star(X,1) + \frac{R_1}{\gamma^\star(X,1)}\left(Y_1 - \mu_1^\star(X,1)\right)\right) - Y_0 - \mu_1^\star(X,0) + \mu_0^\star(X,0)\right)$$
$$- \frac{(1-A)\pi^\star(X)}{(1-\pi^\star(X))\mathbb{E}\left[A\right]}\left(\frac{R_1}{\gamma^\star(X,0)}\left(Y_1 - \mu_1^\star(X,0)\right) - Y_0 + \mu_0^\star(X,0)\right) - \frac{A}{\mathbb{E}\left[A\right]}\theta^\star. \tag{44}$$

*Under Assumption 2.1 and Assumption D.2, the efficient observed-data influence function is given by:*

$$\varphi\left(\mathcal{D}\right) = \frac{A}{\mathbb{E}\left[A\right]}\left(\left(\mu_1^\star(X,Y_0,1) + \frac{R_1}{\gamma^\star(X,Y_0,1)}\left(Y_1 - \mu_1^\star(X,Y_0,1)\right)\right) - Y_0 - \eta_1^\star(X,0) + \mu_0^\star(X,0)\right)$$
$$- \frac{(1-A)\pi^\star(X)}{(1-\pi^\star(X))\mathbb{E}\left[A\right]}\left(\left(\mu_1^\star(X,Y_0,0) + \frac{R_1}{\gamma^\star(X,Y_0,0)}\left(Y_1 - \mu_1^\star(X,Y_0,0)\right)\right) - Y_0 - \eta_1^\star(X,0) + \mu_0^\star(X,0)\right)$$
$$- \frac{A}{\mathbb{E}\left[A\right]}\theta^\star.$$

$$\tag{45}$$

*The semiparametric efficiency bound is given by* $\mathbb{V}\left[\varphi\left(\mathcal{D}\right)\right]$.

*Proof.* The structure of the proof is the same as in 2.8. $\qquad\square$

## D.2 WHEN PRE-TREATMENT OUTCOME CAN BE MISSING

**Assumption D.5** (Missing at random). Let the following MAR assumptions hold:

  **a. No unmeasured confounding.** $Y_t \perp\!\!\!\perp R_t \mid X, A$ for $t \in \{0,1\}$.

  **b. Weak overlap.** $\gamma_t^\star(x,a) = \mathbb{P}\left[R_t = 1 \mid X = x, A = a\right] \in (0,1)$ almost surely for every $x \in \mathbb{R}^p$, $a \in \{0,1\}$ and $t \in \{0,1\}$.

Equipped with the previous sets of assumptions, we can now identify the ATT as a function of the observed data as shown in the following Lemma.

**Lemma D.6** (Identification of ATT). *Under Assumption 2.1 and Assumption D.5, the ATT can be identified as a function of the observed data:*

$$\theta^\star = \mathbb{E}\left[\frac{A}{\mathbb{E}\left[A\right]}\left(\mathbb{E}\left[Y_1 \mid X, A=1, R_1=1\right] - \mathbb{E}\left[Y_0 \mid X, A=1, R_0=1\right]\right)\right]$$
$$- \mathbb{E}\left[\frac{A}{\mathbb{E}\left[A\right]}\left(\mathbb{E}\left[Y_1 \mid X, A=0, R_1=1\right] - \mathbb{E}\left[Y_0 \mid X, A=0, R_0=1\right]\right)\right]. \tag{46}$$

*Proof.* Under Assumption 2.1 and Assumption D.5, the following chain of equalities holds:

$$\theta^\star = \mathbb{E}\left[Y_1^{(1)} - Y_1^{(0)} \mid A=1\right]$$
$$= \mathbb{E}\left[Y_1^{(1)} - Y_0 \mid A=1\right] - \mathbb{E}\left[Y_1^{(0)} - Y_0 \mid A=1\right]$$
$$= \mathbb{E}\left[\mathbb{E}\left[Y_1^{(1)} \mid X, A=1\right] - \mathbb{E}\left[Y_0 \mid X, A=1\right] \mid A=1\right] - \mathbb{E}\left[\mathbb{E}\left[Y_1^{(0)} - Y_0 \mid X, A=1\right] \mid A=1\right]$$
$$= \mathbb{E}\left[\mathbb{E}\left[Y_1^{(1)} \mid X, A=1\right] - \mathbb{E}\left[Y_0 \mid X, A=1\right] \mid A=1\right] - \mathbb{E}\left[\mathbb{E}\left[Y_1^{(0)} - Y_0 \mid X, A=0\right] \mid A=1\right]$$
$$= \mathbb{E}\left[\mathbb{E}\left[Y_1 \mid X, A=1, R_1=1\right] \mid A=1\right] - \mathbb{E}\left[\mathbb{E}\left[Y_0 \mid X, A=1, R_0=1\right] \mid A=1\right]$$
$$- \mathbb{E}\left[\mathbb{E}\left[Y_1 \mid X, A=0, R_1=1\right] \mid A=1\right] + \mathbb{E}\left[\mathbb{E}\left[Y_0 \mid X, A=0, R_0=1\right] \mid A=1\right]$$
$$= \mathbb{E}\left[\frac{A}{\mathbb{E}\left[A\right]}\left(\mathbb{E}\left[Y_1 \mid X, A=1, R_1=1\right] - \mathbb{E}\left[Y_0 \mid X, A=1, R_0=1\right]\right)\right]$$
$$- \mathbb{E}\left[\frac{A}{\mathbb{E}\left[A\right]}\left(\mathbb{E}\left[Y_1 \mid X, A=0, R_1=1\right] - \mathbb{E}\left[Y_0 \mid X, A=0, R_0=1\right]\right)\right]. \tag{47}$$

□

We can also derive the efficient influence function for the previous identified target. Its variance defines the semiparametric efficiency bound.

**Proposition D.7** (Semiparametric efficiency bounds). *Under Assumption 2.1 and Assumption D.5, the efficient observed-data influence function is given by:*

$$\varphi(\mathcal{D}) = \frac{A}{\mathbb{E}[A]} \left( \left( \mu_1^\star(X,1) + \frac{R_1}{\gamma_1^\star(X,1)} (Y_1 - \mu_1^\star(X,1)) \right) - \left( \mu_0^\star(X,1) + \frac{R_0}{\gamma_0^\star(X,1)} (Y_0 - \mu_0^\star(X,1)) \right) \right)$$

$$- \frac{A}{\mathbb{E}[A]} (\mu_1^\star(X,0) - \mu_0^\star(X,0))$$

$$- \frac{(1-A)\pi^\star(X)}{(1-\pi^\star(X))\mathbb{E}[A]} \left( \frac{R_1}{\gamma_1^\star(X,0)} (Y_1 - \mu_1^\star(X,0)) - \frac{R_0}{\gamma_0^\star(X,0)} (Y_0 - \mu_0^\star(X,0)) \right) - \frac{A}{\mathbb{E}[A]} \theta^\star .$$
(48)

*The semiparametric efficiency bound is given by* $\mathbb{V}[\varphi(\mathcal{D})]$.

*Proof.* The structure of the proof is the same as in 2.8. □

# E  REAL-DATA EXAMPLE

To address the practical utility and empirical validation of our estimators, we provide a real-world application using the LaLonde (1986) dataset. This dataset is a canonical benchmark in the causal inference literature, as it allows for the comparison of non-experimental estimators against a known, credible Average Treatment Effect on the Treated (ATT) derived from a randomized controlled trial (Imbens & Xu, 2024).

## E.1  SETUP

The original Lalonde dataset analyzes the effect of the National Supported Work (NSW) job training program on post-treatment earnings. We use the experimental benchmark ATT, $1794.34, as the *ground truth* for our comparison. To simulate the exact problem our paper addresses, we utilize a version of the Lalonde dataset where the pre-treatment outcome (1974 earnings, $re74$) is only partially observed. This specific dataset, which is publicly available, was created by Yang et al. (2023a). This setup creates a challenging and realistic scenario where simply discarding observations with missing pre-treatment data – a *complete-case* analysis – is expected to introduce significant selection bias.

We estimate the ATT of the training program on 1978 earnings ($re78$) using our two proposed estimators (under Assumptions 2.3 and 2.4). We compare their performance against three common baseline estimators:

- Difference-in-Means. A naive comparison of mean $re78$ between the treated and control groups.

- Difference-in-Differences (complete-cases). The standard DiD estimator applied only to the subset of the sample where $re74$ is observed.

- `DR-DiD` (complete-cases): The doubly-robust DiD estimator (Sant'Anna & Zhao, 2020) also applied only to the complete-case samples.

For all estimators, nuisance functions (where applicable) are estimated using Random Forests classifiers and regressors.

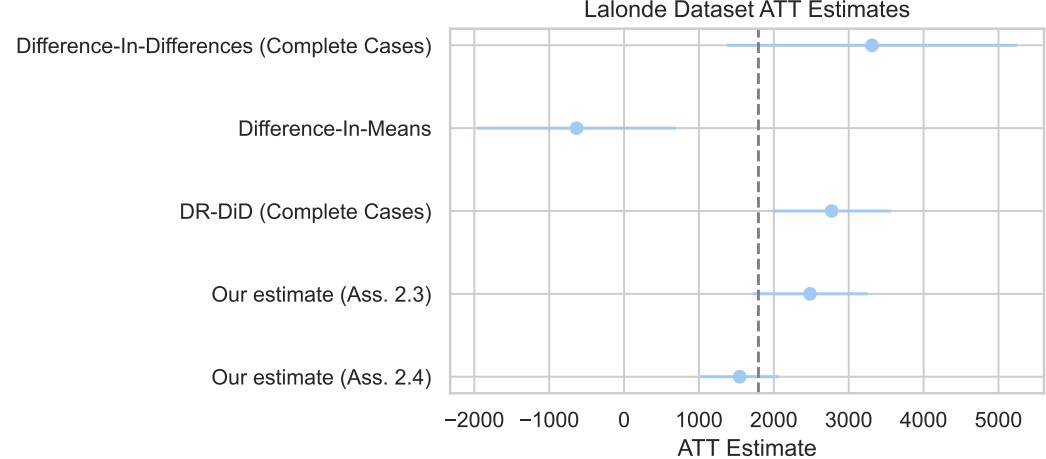

Figure 7: ATT estimates on the LaLonde (1986) dataset with MAR pre-treatment outcomes, as provided by Yang et al. (2023a). The plot compares point estimates and 95% confidence intervals for our proposed estimators (under Assumptions 2.3 and 2.4) against three baselines (Difference-in-Means, complete-case DiD, and complete-case DR-DiD). The dashed vertical line indicates the experimental benchmark ATT ($1794.34). While the baseline and complete-case methods are severely biased, our estimates provide a more principled assessment of the causal effect of the training program on participants' earnings.

### E.2 RESULTS

The results of this empirical application are summarized in Figure 7. The plot clearly demonstrates the practical failure of naive and complete-case methods in this MAR setting. In fact, the Difference-in-Means estimator is severely biased, yielding a negative and statistically insignificant estimate (-$635.03). Estimates relying on a complete-case analysis are also highly biased and misleading. The standard DiD ($3311.02) and the DR-DiD ($2771.82) both substantially overestimate the true effect. Critically, the 95% confidence intervals for the Difference-in-Means and the DR-DiD estimators fail to cover the true experimental benchmark of $1794.34.

In sharp contrast, our proposed estimators, which are designed to handle missing data as arising in this challenge, perform well. Under Assumption 2.3, our estimate is $2483.18; under Assumption 2.4, our estimate is $1544.70. Both estimates are closer to the experimental benchmark ATT. Moreover, the 95% confidence interval of our estimators, both under Ass. 2.3 and Ass. 2.4, successfully contains the experimental benchmark.

This empirical application validates the practical importance of our framework. It shows that by correctly and efficiently accounting for the missing data mechanism, our estimators can correct for the severe selection bias that invalidates standard methods, thereby recovering a credible estimate of the true causal effect.

### E.3 FURTHER ROBUSTNESS COMPARISONS

To further probe the robustness properties of our proposed estimators, we conduct an additional analysis on the LaLonde (1986) dataset. We compare the performance of our estimators against singly-robust variants:

- IPW (Inverse Probability Weighting). Estimators that deal with missing observations in $Y_0$ relying only on the missingness ($\gamma^\star$) models.

- OR (Outcome Regression). Estimators that deal with missing observations in $Y_0$ relying only on the outcome regression ($\mu^\star$) models.

- DR (Doubly Robust). Our proposed estimators, which utilize both $\gamma^\star$ and $\mu^\star$ nuisance functions.

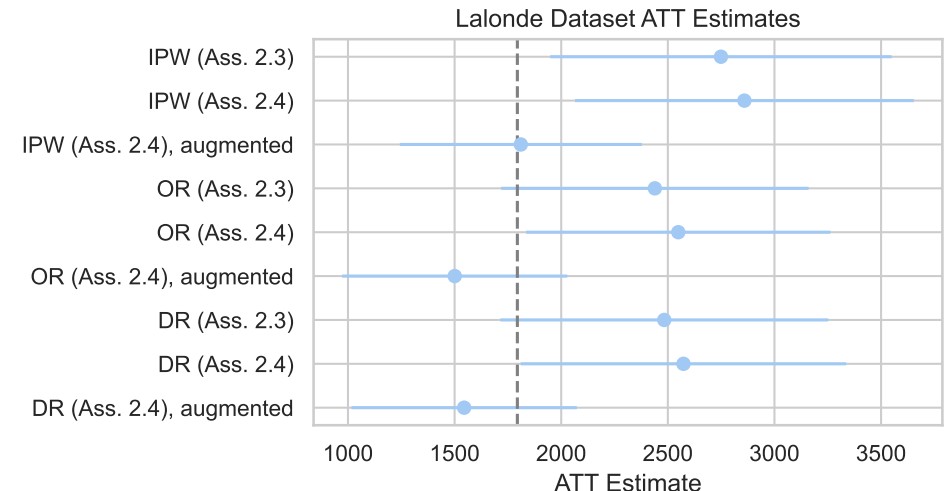

Figure 8: Comparison of singly-robust and doubly-robust ATT estimates on the Lalonde dataset. The plot shows estimates from IPW-only (relying on $\gamma^\star$), OR-only (relying on $\mu^\star$), and DR (relying on both) estimators. *Augmented* refers to the estimation of the nested regression $\eta^\star$ using the method in Theorem 3.4.

Furthermore, for estimators under Assumption 2.4, we compare standard non-augmented versions against the *augmented* versions. The non-augmented estimators learn the nested regression $\eta^\star$ using only the outcome model $\hat{\mu}$. The augmented estimators, by contrast, use the strategy from Theorem 3.4, incorporating both $\hat{\mu}$ and $\hat{\gamma}$ to estimate $\eta^\star$.

The results, shown in Figure 8, are striking. All non-augmented estimators are overestimating the target. Moreover, 95% confidence intervals based on the non-augmented IPW estimators fail to cover the \$1794.34 experimental benchmark, as well as non-augmented estimators based on Assumption 2.4. All the other estimators provide instead correct coverage. Figure 8 highlight the usefulness of our proposed augmentation strategy. While the non-augmented estimators for Assumption 2.4 present some bias, the augmented versions of all three estimators – IPW, OR, and DR – perform exceptionally well. All three point estimates are clustered closely around the true benchmark, and their confidence intervals cover the true value. This powerfully illustrates the practical value of the augmentation described in Theorem 3.4. In a challenging, real-world scenario where all nuisance function models are unknown, the augmentation provides crucial protection against bias and is essential for recovering a credible causal estimate.

