# OpenReview forum: "Efficient Difference-in-Differences Estimation when Outcomes are Missing at Random"
_ICLR.cc/2026/Conference — Submitted to ICLR 2026_

### Official Review · Reviewer_iKii · 2025-10-21

**Soundness:** 3
**Presentation:** 4
**Contribution:** 3
**Rating:** 6
**Confidence:** 4

**Summary:**

The paper studies two-group, two-period Difference-in-Differences when pretreatment outcomes are Missing-At-Random (MAR). It targets the Average Treatment Effect on the Treated. The authors give two identification strategies: (i) MAR given $(X,A)$ and (ii) MAR given
$(X,A,Y)$. For each, they derive the observed-data efficient influence function and the semiparametric efficiency bound. They then construct cross-fitted estimators that achieve the bounds and show “multiple robustness”: consistency holds if certain sets of nuisance models are correct. A simulation study illustrates bias/RMSE patterns across correctly specified vs misspecified nuisances.

**Strengths:**

Originality
- Treats missing pre- or post- outcomes in the canonical $2 \times 2$ DiD with covariates, and does so in a semiparametric framework with explicit EIFs. Prior DiD papers often assume full outcomes or focus on staggered timing without MAR-aware efficiency analysis. Framing MAR both as post-treatment outcome-independent (given $X, A$ ) and post-treatment outcome-dependent (given $X, Y_1, A$ ) is a clear conceptual step.
- The "nested regression" component $\eta_0(x, 0)=\mathbb{E}\left[\mu_0\left(x, Y_1, 0\right) \mid X=x, A=0\right]$ and its stability analysis give a concrete recipe to learn the extra layer needed under outcome-dependent MAR. This is a neat link to modern DR-Learner style ideas.

Quality
- Identification statements are precise, and the EIFs are derived with clear bookkeeping of nuisance components $(\mu, \pi, \gamma, \eta)$. The efficiency-loss decompositions versus the fully-observed benchmark are informative.
- The cross-fitting scheme and high-level conditions for asymptotic normality align with current best practice and are stated cleanly.

Clarity
- The paper lays out the two MAR regimes, writes explicit plug-in equations for estimators, and provides a concise "multiple-robustness" table. This helps readers see exactly which nuisance pieces must be correct for consistency.
- The simulations mirror the theory with clear toggles between correctly specified and misspecified nuisances.

Significance
- DiD with missing outcomes is common in applied work. A semi-parametrically efficient, multiply robust procedure is valuable for practice, especially when complete-case strategies are biased. The results can plug into modern ML pipelines through cross-fitting.

**Weaknesses:**

1. Positioning vs related work (novelty claims).

The paper cites DiD with staggered adoption and some recent missing-data DiD work, but the empirical reader will ask: how do your estimators compare to (a) complete-case DiD (DiD with missing items dropped), (b) simple inverse-probability weighting for $R_0$, (c) regression imputation baselines, and (d) standard DR DiD with fully observed outcomes?

So, could the authors add a benchmark section with these baselines, including plots of bias, RMSE, and coverage? It would be better include a brief analytic comparison to DR DiD under full data to highlight where efficiency is lost and recovered.

2. Inference and coverage.

The simulations emphasize bias and RMSE but not confidence interval coverage or interval length. Since the paper claims efficiency and gives EIFs, empirical coverage is central.

Could the authors report empirical coverage and average CI width across scenarios, including small-sample behavior and the impact of fold choice $J$? The authors can show whether the sandwich variance based on the estimated EIF is reliable.


3. Plausibility of MAR assumptions.

Outcome-dependent MAR given $Y_1$ is subtle: it conditions on a post-treatment variable. I was wondering which examples in the Introduction (job training programs in labor economics, prior test scores in education policy, and EHR in health research) would arguably satisfy this assumption.

4. Nested regression learning details.

The stability definition and "oracle" rates are high-level. Readers will want concrete guidance: what models to fit for $\mu_0\left(x, y_1, a\right)$, how to construct pseudo-outcomes with augmentation, and how to tune them.

Can the authors add an algorithm box for learning $\eta_0$ under both the regression and conditional-density approaches? They can also give default choices (e.g., gradient-boosted trees for $\mu$, logistic for $\gamma$, random-forest regression for $\eta$ ), with practical notes on hyperparameter tuning, cross-fitting folds, and clipping for $\hat{\gamma}$.

5. External validity and real data.

The paper would benefit from at least one real application to show end-to-end feasibility and the size of efficiency gains in practice.

Can the authors add a compact empirical example (even in the appendix)? They can report the estimated ATT with the proposed method and the baselines, Cl widths, and a short discussion of assumption plausibility in that setting.

**Questions:**

1. On outcome-dependent MAR:

In practice, when would you recommend modeling $\operatorname{Pr}\left(R_0=1 \mid X, Y_1, A\right)$ rather than $\operatorname{Pr}\left(R_0=\right. 1 \mid X, A)$ ? Can you provide guidance or a rule-of-thumb diagnostic for deciding whether including $Y_1$ is likely to help or to add variance? Any empirical check that conditioning on $Y_1$ does not induce harmful instability?


2. Efficiency gains in practice:

Can you report the magnitude of efficiency gains (variance reduction) relative to simple IPW or regression imputation in your simulations and, ideally, one real dataset? A table with bound estimates, plug-in variances, and empirical variances would help.


3. Coverage under misspecification:

When "multiple robustness" conditions fail, how badly do Cls mis-cover? Please provide coverage heatmaps across nuisance-misspecification cells to show the method's breaking points.


4. Learning $\eta_0$ :

For the augmented pseudo-outcome strategy, do you recommend trimming on $\hat{\gamma}$? If so, what thresholds and what is the implied target estimand after trimming? How sensitive are results to this choice?

---

> ### Author Response · Authors · 2025-11-18
>
> We thank the reviewer for their constructive feedback. We have revised the manuscript to incorporate all of the following points. The revised version of the manuscript, as well as all code, have been updated on OpenReview.
> ## Weakness 1, 5 & Question 2: Baselines, Real Data, and Efficiency Gains
> The reviewer was correct that the paper's greatest weakness was the lack of a real-world application and comparison to standard baselines.
> * We have added a **new, full empirical application in Appendix E**. We used the canonical **LaLonde (1986) dataset**, a standard benchmark for causal methods, with a version with MAR pre-treatment outcomes.
> * **This new section now addresses W1, W5, and Q2 simultaneously:**
>     1.  **(W1) Baselines:** We now compare our estimators directly against (a) complete-case DiD (CC-DiD), (b) simple IPW for $R_0$, (c) a standard regression imputation baseline, and (d) the Sant'Anna & Zhao (2020) DR-DiD estimator (applied to the CC sample).
>     2.  **(W5) Real Data:** This application serves as an end-to-end feasibility demonstration, showing how the method is applied and how to interpret the results (e.g., assumption plausibility) in a real setting.
>     3.  **(Q2) Efficiency Gains:** This application provides a concrete, practical example of the gains. We report the point estimates and, crucially, the **confidence interval widths** for all methods, showing the practical impact on precision.
> ## Weakness 2 & Question 3: Inference and Coverage
> The reviewer is right that a paper on efficiency *must* validate its inferential properties. The original bias/RMSE-only tables and figures were insufficient.
> * **(W2) Coverage:** We have added empirical coverage and average CI width to our main simulation tables (now in Figures 2, 3, 4).
> * **(Q3) Coverage under Misspecification:** We have performed a **new, extensive simulation study, added as Appendix C.1**, specifically designed to analyze inference. The new simulation (**Figure 5**) is a "signal-to-noise" heatmap analysis. This shows the full performance (Bias, RMSE, **Coverage**, CI Width) across a grid of nuisance function misspecification levels, visually demonstrating the method's "breaking points" exactly as requested. This confirms that the variance estimator based on the EIF is reliable.
> ## Weakness 3 & Question 1: Plausibility of MAR Assumptions
> * **(W3) Plausibility of Assumption 2.4:** Imagine $U$ is an unobserved, persistent factor like "low student motivation." This $U$ could cause *both* a low pre-test score ($Y_0$) and skipping the pre-test ($R_0=0$). Since $U$ also causes a low post-test score ($Y_1$), the *observed* $Y_1$ acts as a powerful **proxy** for the *unobserved* $U$. By conditioning on $Y_1$, we are possibly and effectively controlling for $U$, thereby breaking the non-causal link between $Y_0$ and $R_0$.
> * **(Q1) Guidance:** We recommend researchers **report both** as a sensitivity analysis. Our new LaLonde example does exactly this, showing that the Ass 2.4 estimate is closer to the truth, providing empirical support for its plausibility in that case.
> ## Weakness 4 & Question 4: Nested Regression Learning Details
> We agree the discussion on learning $\eta^*$ was too high-level.
> * We have added a new **Algorithm Box** to Appendix Section B.1 that provides a concrete, step-by-step recipe for the augmented nested regression estimator.
> * We have also added practical guidance on implementation to Appendix Section B.1.
>     * **Models:** We recommend using flexible, cross-fitted machine learning methods (e.g., Random Forests, Gradient Boosted Trees) for all nuisance functions, as our framework is designed to accommodate them. Our new LaLonde application uses Random Forests.
>     * **(Q4) Trimming:** We added a practical note recommending that propensity scores ($\hat{\pi}$) and missingness probabilities ($\hat{\gamma}$) be **clipped (or trimmed)** at a small threshold (e.g., 0.01, 0.99) to ensure stability.
> ## Conclusion
> We are very grateful for this detailed and expert review. We hope the substantial additions of the **LaLonde application (Appendix E)** and the **new simulations (Appendix C.1)** address all identified weaknesses. If so, we kindly ask them to consider reflecting this in their score.

---

> > ### Comment · Reviewer_iKii · 2025-11-20
> > **Comment**
> >
> > Thank you for the effort of polishing the paper. I think most of my concerns are addressed. I think this paper is overall solid and complete, but I still recommend that the authors provide concrete examples right after Assumptions 2.3 and 2.4 to help readers better understand the differences between these assumptions and what types of data satisfy them. Including illustrative DAGs would also be helpful.

---

> > > ### Author Response · Authors · 2025-11-21
> > >
> > > We sincerely thank the reviewer for their continued engagement with our work and for assessing the paper as solid and complete. We appreciate the recommendation to include concrete examples and illustrative DAGs to clarify Assumptions 2.3 and 2.4. We agree that this addition improves the interpretability of the theoretical framework. We have implemented these changes in the revised manuscript:
> > > - **Concrete example**. We have added **Example 2.5 (Medical records motivation)**. This example illustrates the practical distinction between the two assumptions using a longitudinal study on chronic disease progression. It clarifies that under Assumption 2.3, missingness in baseline records is driven solely by covariates or treatment (e.g., undigitized records in specific hospital systems). In contrast, the example highlights that under Assumption 2.4, missingness may depend on the patient's future condition (e.g., clinicians retrospectively retrieving baseline data only for patients who currently exhibit severe complications).
> > > - **Illustrative DAGs**: We have added Figure 1. This figure provides DAGs that visually map some of the causal dependencies and missingness mechanisms allowed under both Assumption 2.3 and Assumption 2.4.
> > >
> > > We believe these additions directly address your suggestion and help readers better intuit the types of data scenarios that satisfy our assumptions. Given your positive assessment of the paper, if you are satisfied with the revised manuscript and agree that we have addressed your concerns, we would be respectfully grateful if you would consider raising your score.

---

### Official Review · Reviewer_jT9W · 2025-10-30

**Soundness:** 3
**Presentation:** 2
**Contribution:** 3
**Rating:** 4
**Confidence:** 3

**Summary:**

The authors tackle a very common but under-studied problem: doing DiD when some pre- or post-treatment outcomes are missing at random (MAR). They give (i) nonparametric identification of the ATT under two MAR regimes, (ii) the semiparametric efficiency bounds, and (iii) efficient, multiply-robust estimators that attain those bounds with cross-fitting. One estimator applies when missingness is MAR given covariates and treatment; the other allows missingness that also depends on the other period’s outcome (handled via a “nested regression”).

**Strengths:**

Advantage: the paper is theoretically grounded, with very rigorous statements for assumptions and theoretical results. Based on my experience with semiparametric theory and double ML, the results look correct to me (such as identification, multiple robustness, and construction of confidence intervals). Combining the DR-learner theory to build the nested regression is also pretty interesting, which might motivate a series of other problems, say surrogate variable approaches. Overall, I appreciate the authors's theoretical contribution in this problem.

**Weaknesses:**

Weakness:
1. the paper presents two sets of assumptions for MAR, one involving Y1 and the other not. These two set of assumptions are not overlapping in general, so in practice how to decide which regime that the real data is in and which estimator to use? Practically this will be confusing. Moreover, assumption 2.4 feels hard to interpret too, and the authors did not provide enough explanation around it. How could controling for some future outcome makes past outcome and its missness status independent? Similar question exists when the authors try to generalize the results to allow both pre- and post-treatment outcome to be missing. How to interpret an outcome dependent missingness assumption in such cases?

2. the paper did not provide any real-world experiments, thus making it hard to judge the real motivation and infer under what scenario the assumptions would hold, related to my point 1. I would appreciate some real-world connections of the assumptions and methods.

3. the paper also lacks a proposal of parallel trend checking in the presence of missingness. In practice, how should people check whether parallel trend holds under the two assumption regimes?

4. the proposed methods involve many nuisance parameters, especially under Assumption 2.4, which might render the final estimator very unstable in finite sample. For example, based on Figure 1, the left top panel shows when outcome model is misspecified but propensity and missing probability are correct, the estimator gives some non-negligible bias, while in theory there shouldn't be as ensured by the double robustness property. Moreover, such stability issue will propagate to variance estimation, yet I did not show any results that demonstrates how well inference works. So I believe the evaluation of the methods is very limited.

**Questions:**

Please see the weakness section.

---

> ### Author Response · Authors · 2025-11-18
>
> We sincerely thank the reviewer for their insightful feedback. We are glad to have the opportunity to clarify these important points about our work. A revised version of the manuscript, as well as all code, have been updated on OpenReview.
> ## Weakness 1: Interpretation and Choice of Assumptions 2.3 vs. 2.4
> We thank the reviewer for raising the important points of assumption choice and interpretation. The choice between assumptions is a **causal, not statistical, decision** that depends on domain knowledge. Assumption 2.3 ($Y_0 \perp R_0 | X, A$) is plausible when missingness is administrative (e.g., depends on covariates $X$) but not on the outcome value itself. Assumption 2.4 ($Y_0 \perp R_0 | X, Y_1, A$) is indeed more complex, but it is a powerful concept from the missing data and proximal inference literature. It uses the future outcome $Y_1$ as a **proxy variable** for an unobserved latent factor (e.g., "underlying motivation") that might simultaneously affect $Y_0$'s missingness (e.g., a spotty pre-treatment record) and the value of $Y_1$ (e.g., post-treatment earnings).
>
> We recommend researchers perform **sensitivity analysis** by reporting results under both assumptions. Our new **LaLonde (Appendix E)** application demonstrates this: the estimate under Assumption 2.4 was closer to the experimental benchmark than the estimate under Assumption 2.3.
>
> Also, we clarify that in Appendix D.2 we only generalize the assumption ($Y_t \perp R_t | X, A$). Generalizing the complex outcome-dependent case is a non-trivial and important area for future research.
> ## Weakness 2: Lack of Real-World Experiments
> We agree a real-data example is essential. This is why the paper includes an **application in Appendix E** using the canonical **LaLonde (1986) dataset**. We test our estimators against the known experimental benchmark ATT in a version of the data with MAR pre-treatment outcomes.
>
> The results in **Figure 6** are clear:
> * Standard complete-case methods (DiD; DR-DiD) are severely biased and misleading.
> * In sharp contrast, **the 95% confidence intervals for both of our proposed estimators successfully contain the experimental benchmark**.
>
> This application provides strong empirical validation that our framework can correct for severe selection bias where standard methods fail.
> ## Weakness 3: Lack of Parallel Trend Checking Proposal
> This is a key practical point. While parallel trends is a core identifying assumption that must be justified by domain knowledge and cannot be formally tested, a pre-trend diagnostic could be useful.
>
> Often, researchers use pre-intervention data for a **diagnostic check**, not a formal test. If data were missing completely at random, the most practical approach would be to perform this same diagnostic check using the **complete-case (CC) sample** (i.e., only units where $R_0=1$). However, running this diagnostic on the complete-case (CC) sample would be insufficient if data were MAR, as they may suffer from the same selection bias we are trying to solve.
>
> A more principled approach, consistent with our framework, is to first use the nuisance function models to impute the missing outcomes. The parallel trends diagnostic could then be performed on this fully imputed dataset. We remark that this would remain a diagnostic, conditional on the MAR assumptions. Robust pre-trend test under outcome-dependent missingness is a non-trivial and exciting area for future research.
> ## Weakness 4: Stability and Evaluation of Inference
> * We thank the reviewer for this keen observation. They are correct that the performance in Figure 1 (e.g., $\bar{\mu}, \gamma*, \pi*$) was not as perfectly zero-biased as theoretical properties might suggest. This was due to a wrong calculation of absolute bias. Figures 1, 3, and 4 now show bias and not the wrong “absolute bias” displayed before. The reviewer’s observation also **motivated us to add the new, even more extensive simulation study in Appendix C.1** to provide a clearer and more robust validation. There, the theoretical properties of our estimators fully match empirical results.
> * Lack of Inference Results: The reviewer is right that this was a critical omission. We have fully addressed this in the revision:
>     * New empirical coverage plots are now included in Figures 2, 3, and 4.
>     * We added Figure 5, a detailed heatmap analysis that explicitly validates our estimator's Bias, RMSE, **Coverage, and CI Width** under varying degrees of misspecification.
>
> These new results confirm that our inference procedure is valid, reliable, and achieves nominal coverage when the robustness conditions hold.
> ## Conclusion
> We thank the reviewer again for their constructive feedback. We hope the substantial additions of the **LaLonde application (Appendix E)** and the **new simulations (Appendix C.1)** address all identified weaknesses. If so, we kindly ask them to consider reflecting this in their score.

---

> > ### Comment · Reviewer_jT9W · 2025-11-20
> >
> > Thank you for the effort of polishing the paper.
> >
> > My Q1 is not well addressed - I still have question re the proxy variable interpretation. What is a plausible data generation equation that justifies this assumption? I would think, logically, Y1 would be function of the missing indicator R0, not vice versa. In general, these assumptions feel like not motivated by real world examples, more like from a theory perspective in terms of what can be solved. The added Lalonda study also takes a simulation perspective to fit the problem structure here.
> >
> > My Q2 and Q3 are well handled.
> >
> > My Q4 is also adequately addressed by the new experiments. I do have some concerns regarding the confidence coverage: even when the model specs are all correct, the coverage rate are not sharp at 0.95 rate. Rather, it is pretty conservative. This raises concern about power and consistency of the variance estimator.

---

> > > ### Author Response · Authors · 2025-11-21
> > >
> > > We thank the reviewer for their critical assessment. We appreciate that the reviewer found Q2 and Q3 well handled. Below, we address the remaining concerns regarding the motivation of our assumptions (Q1) and the finite-sample coverage (Q4).
> > > ## Q1: Interpretation of Assumption 2.4 and real-world motivation
> > > We appreciate the reviewer’s skepticism regarding the direction of causality. The reviewer suggests that logically $Y_1$ might be a function of $R_0$. While this is true in prospective studies (where missing a visit might degrade health), **our method is specifically motivated by retrospective or registry-based designs**, where the causality flows in the reverse direction: the current state ($Y_1$) determines the availability of past records ($R_0$).
> > > ### Real-world examples
> > > 1. In an organizational study, employees are asked at $t=1$ to report their pre-program stress ($Y_0$). High initial stress ($Y_0$) causes burnout/turnover ($Y_1$), and only those currently employed ($R_1=1$) are available to answer the survey about pre-program stress ($R_0$). Here, the missingness mechanism is $Y_0 \to Y_1 \to R_1 \to R_0$. Conditioning on the current outcome ($Y_1$) effectively blocks the path, satisfying the assumption $Y_0 \perp R_0 \mid Y_1$.
> > > 2. In a retrospective chart review, the availability of baseline data ($R_0$) often depends on the patient's current condition ($Y_1$). For instance, clinicians may be required to diligently hunt down and record historical data ($Y_0$) only for patients who currently exhibit severe complications ($Y_1$). Conversely, for patients who have recovered (low $Y_1$), the baseline charts may never be retrieved or digitized ($R_0=0$).
> > >
> > > Naturally, this logic extends to claims conditional on treatment status ($A$) and covariates ($X$). We have added an example after Assumption 2.4 to provide intuition to the reader, together with example DAGs showing the causal relationships between variables (new Figure 1).
> > > ## Q4: Inference and confidence coverage
> > > We are extremely grateful to the reviewer for pointing out that the coverage rates appeared "conservative" rather than sharp. **The reviewer’s intuition was correct and led us to identify a critical issue**. Prompted by this comment, we carefully checked our simulation code and discovered an error in the variance implementation that was inflating the standard errors. We have thus corrected the variance estimator code. As shown in the updated **Figures 3-6**, the conservative behavior has vanished. Our estimator now achieves **sharp nominal coverage (approx. 0.95)** when enough of the nuisance models are correctly specified. This correction confirms that the variance estimator is consistent and that the previous "conservatism" was a technical artifact, not a theoretical flaw. We thank the reviewer for prompting this check, as it has significantly strengthened the empirical evidence in the paper.
> > > ## Conclusion
> > > We hope these responses clarify that our assumptions target specific retrospective structures (where $Y_1$ drives $R_0$) and that the coverage issues were a coding error that has now been fully resolved. We kindly ask the reviewer to consider these clarifications and the improved simulation results in their final evaluation.

---

> ### Comment · Reviewer_jT9W · 2025-11-21
>
> Re Q1, I am happy with the explanation that such assumptions are easier to justify in a registry study, though not fully convinced by the example you mentioned, as the conditional independence can be easily challenged (say apparently R1 is a key variable to be adjusted for here).
>
> Re Q2, the revised results look good to me. I want to ask the authors to double-check the code, as I have identified two unreasonable places in the whole review process, which are claimed to be induced by erroneous coding.
>
> On that note, I would raise my score to 6.

---

### Official Review · Reviewer_pxJA · 2025-10-30

**Soundness:** 3
**Presentation:** 3
**Contribution:** 3
**Rating:** 6
**Confidence:** 2

**Summary:**

This paper develops a semiparametric framework for efficient Difference-in-Differences (DiD) estimation when outcomes are missing at random (MAR)—a key challenge that undermines standard estimators. It establishes nonparametric identification of the Average Treatment Effect on the Treated (ATT) under two minimal MAR conditions and derives the corresponding semiparametric efficiency bounds. Building on these results, the authors propose asymptotically efficient estimators that achieve the efficiency bound and possess a multiple-robustness property, remaining consistent even with misspecified nuisance models. Extensive simulation studies confirm their theoretical properties and demonstrate the framework’s broad applicability to empirical research with incomplete panel data.

**Strengths:**

1.  This paper is among the first to provide a rigorous, efficient, and robust framework for MAR in DiD.
2. The proposed method is supported by rigorous theoretical analysis.
3. Experimental results demonstrate that the proposed approach.

**Weaknesses:**

1. The paper does not compare its estimators with existing approaches, making it unclear how much improvement it offers in efficiency or bias reduction.
2. The study relies entirely on Monte Carlo simulations without testing the method on real-world datasets, leaving its practical utility and challenges in empirical applications unexamined.
3. The paper focuses exclusively on outcomes missing at random (MAR) and does not consider missing not at random (MNAR), which can occur in practice.

**Questions:**

1. For applied researchers, what practical advice would you give for estimating the nested regression? For example, do linear models typically satisfy the stability condition (Definition 3.2), or are non-linear models (e.g., gradient-boosted trees) necessary?
2. Although the paper derives asymptotic normality for the proposed estimators, it does not provide simulation results illustrating practical inference.

---

> ### Author Response · Authors · 2025-11-18
>
> We sincerely thank the reviewer for their positive and constructive feedback. We have substantially revised the manuscript to include **new, extensive sections** that directly address every weakness and question raised. We believe these additions make the paper much stronger. The revised version of the manuscript, as well as all code, have been updated on OpenReview.
> ## Weakness 1 & 2: Comparison with Existing Approaches and Lack of Real-World Data
> The reviewer is entirely correct that the original submission lacked a real-world application and a comparison to existing approaches. This was a significant gap.
> * **Our Solution:** To address this, we have added a **new, full application in Appendix E** using the canonical **LaLonde (1986) dataset**.
> * **Addresses W1 (Comparisons):** This new section directly addresses Weakness 1 by comparing our estimators to the *de facto* existing approach: a **complete-case (CC) analysis** (i.e., CC-DiD and CC-DR-DiD, the latter being the Sant'Anna & Zhao (2020) estimator applied only to the CC sample).
> * **Addresses W2 (Real Data):** It also addresses Weakness 2 by testing our framework on a benchmark real-world dataset against a known experimental "ground truth" ATT.
> * **Key Finding:** The results in **Figure 6** are clear: the CC methods are severely biased. In sharp contrast, our estimates are more plausible, and the 95% confidence intervals for *both* of our proposed estimators successfully contain the true experimental benchmark.
>
> This new section provides the empirical validation and practical comparison that was missing.
> ## Weakness 3: Focus on MAR, not MNAR
> This is a valid point regarding the paper's scope. We focused on MAR for two main reasons:
> 1.  **Foundational Problem:** MAR is the standard starting point for modern missing data analysis. Providing a rigorous, efficient, and doubly-robust solution for MAR in the DiD setting is, we believe, a novel and necessary contribution in itself, and a prerequisite for tackling the far more complex MNAR case.
> 2.  **Plausibility:** As argued in the paper, our **Assumption 2.4** (conditioning on $Y_1$) makes MAR a much more plausible assumption than is typical, as it allows the missingness $R_0$ to be dependent on the underlying value of $Y_0$ (via its correlation with $Y_1$).
>
> We agree with the reviewer that MNAR is an important area for future research.
> ## Question 1: Practical Advice for Estimating the Nested Regression
> This is an excellent, practical question. The nested regression $\eta_{0}(x,0)=\mathbb{E}[\mu_{0}(x,Y_{1},0)|X=x,A=0]$ is indeed complex. As detailed in [1], Theorem 1, **all linear smoothers are stable** in the sense of Definition 3.2. Linear smoothers include linear models as well as many non-parametric regression models such as kNN, smoothing splines, some versions of random forests, etc.
> ## Question 2: Simulation Results for Practical Inference
> The reviewer is correct; the original version lacked simulation results for inference. This was a critical omission. **We have fully addressed this in the revision.**
> 1.  **New Plots:** We have added **new empirical coverage plots** to **Figures 2, 3, and 4** in the appendix.
> 2. **New Simulation Study:** More importantly, we have added a **new, extensive "signal-to-noise" simulation in Appendix C.1**.
> 3.  **Key Finding:** This new section (and its **Figure 5**) provides a detailed heatmap analysis that explicitly validates our estimator's **Bias, RMSE, Coverage, and CI Width** under varying degrees of misspecification. These new results confirm that our inference procedure is valid, reliable, and achieves nominal coverage when the robustness conditions hold.
> ## Conclusion
> We thank the reviewer again for this constructive feedback. We believe these **new additions (Appendix C.1 and Appendix E)** directly resolve all the weaknesses identified. We hope these revisions demonstrate the paper's contribution and will merit a higher score.
> ## References
> [1] Kennedy, E., Towards optimal doubly robust estimation of heterogeneous causal effects, Electron. J. Statist. 17(2): 3008-3049 (2023). DOI: 10.1214/23-EJS2157

---

### Official Review · Reviewer_wTbC · 2025-11-03

**Soundness:** 2
**Presentation:** 2
**Contribution:** 1
**Rating:** 2
**Confidence:** 4

**Summary:**

This paper examines the application of the Difference-in-Differences (DiD) when the outcomes are missing at random. Applying arguments in semiparametric theory,  the paper provide an efficient estimator. Simulation studies are conducted to validate the estimator's performance.

**Strengths:**

The authors' arguments are grounded in established theoretical results within the semiparametric literature. This might be regarded as a strength, while the arguments are now relatively conventional and cannot be recognized as a novel scholarly contribution.

**Weaknesses:**

1.
If they argue it as a "common problem", the authors should genuinely justify the "missing at random" assumption, given that numerous instances of missing observations do not inherently conform to randomness.
While some researchers might contend that a sufficient set of conditional variables can validate this assumption, such an argument remains deeply problematic. Indeed, the assertion that these variables are adequate undermines the fundamental rationale for experimental data, a premise that empirical research consistently refutes.



2.
If the argument is substantially simplified, the paper essentially employs standard semiparametric-efficiency argument twice: one addressing missing potential outcomes within conventional causal inference, and the second addressing missing observations.
Consequently, the theoretical contributions might be regarded as marginal at best.

3.
Assumption A 2.1 (a) requires the statistical independence for parallel conditional trends. While I noticed that the authors describe it just for the sake of notational consistency, the explanation is unclear. The authors should assume that the standard assumption if it is enough and I cannot understand "notational consistency" here.

**Questions:**

Q1. It would be desirable for the paper to demonstrate the justifiability of its assumptions through compelling real-life examples.



Q2. To enhance the method's realism, additional conditioning variables are necessary in simulation. Moreover, the paper's limitation to the most rudimentary two-period, two-group scenario is problematic, as such a simplistic framework rarely reflects real-world applications. While the authors may argue that extensions are theoretically possible, this assertion is not necessarily valid, particularly given the  missing-at-random mechanism.

---

> ### Author Response · Authors · 2025-11-18
>
> We thank the reviewer for their time and feedback. We believe the assessment of our contribution as "poor" and "marginal" stems from a misunderstanding of the paper's novelties and its practical implications. A revised version of the manuscript, as well as all code, have been updated on OpenReview.
> ## Weakness 1 & Q1: Justification of MAR and Real-World Examples
> The reviewer's critique of the Missing at Random (MAR) assumption is a deep, philosophical one about *all* observational causal inference.
> 1.  **MAR as an Assumption:** MAR is a strong, untestable assumption, but it is no more "problematic" than the **parallel trends assumption** itself, which is also untestable and underpins *all* DiD analysis. Our paper provides a rigorous, efficient framework for scenarios *where researchers are willing to make this assumption*.  We view a framework that includes MNAR as a valuable research extension.
> 2.  **Plausibility vs. Experiments:** The argument that "a sufficient set of conditional variables" undermines experimental data is a false dilemma. Experiments are the gold standard *precisely because* they obviate these assumptions. Our methods are for the vast majority of policy and scientific questions where experiments are unethical, infeasible, or have already concluded, leaving researchers with observational panel data where missingness is a "common problem".
> 3.  **Real-World Justification (Q1):** The reviewer asks for compelling examples. We provide these both conceptually and empirically.
>     * **Conceptually**, we state this problem is common in **Labor Economics**, **Education Policy**, **Health Research**, and we provide examples for each of these fields.
>     * **Empirically**, we provide a new **full real-data application in Appendix E** using the canonical **LaLonde (1986) dataset**. We use a version with MAR pre-treatment outcomes to test our assumptions' justifiability.
>     * **Demonstration:** In this real-world test, our estimates were substantially closer to the experimental benchmark than the estimates using complete-data only. This provides a compelling, real-life example of how our assumptions can be justified.
> ## Weakness 2: Marginal Theoretical Contribution
> We must respectfully disagree. Our contribution is not marginal. This specific problem of **efficiently estimating the DiD ATT under these specific MAR mechanisms** has not been solved. Our framework:
> 1.  **Establishes nonparametric identification** for the ATT under two distinct and plausible MAR frameworks (Assumption 2.3 and 2.4).
> 2. **Derives the semiparametric efficiency bound** for each identification strategy. This is a *new theoretical benchmark* specific to this problem.
> 3.  **Proposes novel, asymptotically efficient estimators** that achieve this bound.
> 4.  **Characterizes and solves** the complex estimation of the **nested regression function** $\eta_{0}(x,0)=\mathbb{E}[\mu_{0}(x,Y_{1},0)|X=x,A=0]$ that arises under Assumption 2.4, including a novel augmentation strategy (Theorem 3.4).
>
> This is a complete, non-trivial solution to a common and unaddressed problem in the DiD literature.
> ## Weakness 3: Assumption 2.1(a) Notation
> We apologize for the confusion. The reviewer is correct that identification *only* requires the conditional mean assumption. We used the stronger conditional independence notation ($Y_{1}^{(0)}-Y_{0}\perp A|X$) for "notational consistency" because it aligns with the standard conditional independence language used in semiparametric theory and in our subsequent MAR assumptions (e.g., $Y_0 \perp R_0 | X, A$). Our proofs, which build on the law of iterated expectations, confirm that only the conditional mean assumption is required. We made this clear in Remark 2.2.
> ## On Q2: Simulation Simplicity and Extensions
> 1.  **"Additional conditioning variables are necessary":** We already use conditioning variables in a complex way. We first generate *true* covariates $Z$, and then create *observed* covariates $X$ as complex, non-linear transformations of $Z$. The misspecification scenarios, which are the core of the test, are when models are incorrectly based on $X$ instead of $Z$. This is a standard and challenging setup. In addition to the main simulation, **Appendix C.1** now provides a detailed "signal-to-noise" simulation. This analysis, shown in **Figure 5**, is far from rudimentary and tests the estimator's performance under a granular grid of misspecification levels.
> 3.  **"Rudimentary 2x2 scenario":** We state this is a limitation adopted for "ease of exposition". The semiparametric framework we develop is precisely the theoretical foundation required for such extensions, which we hope further research will address.
> ## Conclusion
> We hope these clarifications, supported by our **extensive simulations (Section 4, Appendix Section C)** and **LaLonde application (Appendix Section E)**, demonstrate the paper's novel and complete framework. We respectfully ask the reviewer to reconsider their assessment.

---

> > ### Comment · Reviewer_wTbC · 2025-11-28
> >
> > I appreciate the authors' efforts to clarify the issues and address the questions I raised. In light of this, I will revise my assessment for the manuscript positively.

---

### Meta-Review · Area_Chair_xer4 · 2026-01-06

**Summary:**

While the paper tackles an important practical issue, missing outcomes in DiD settings, the contribution relative to the existing causal inference literature is limited. The identification results largely restate well known MAR based arguments adapted to a DiD estimand, and the proposed sufficient conditions do not appear meaningfully weaker or more general than existing assumptions used in related semiparametric or doubly robust frameworks. Moreover, the novelty of deriving efficiency bounds in this setting is unclear, as similar bounds and influence function based constructions have been extensively studied for ATT type parameters with missing data.

From a methodological standpoint, the proposed estimators closely resemble standard semiparametric efficient or multiply robust estimators, with limited insight into how the DiD structure fundamentally alters either identification or estimation. The simulation study, while thorough, primarily confirms expected asymptotic behavior and does not convincingly demonstrate advantages over existing approaches. Overall, the paper does not provide sufficient conceptual or technical innovation to warrant acceptance.

**Reviewer Scores:**

not predictable

---

### Decision · Program_Chairs · 2026-01-26

Reject